# LRR: Language-Driven Resamplable Continuous Representation against Adversarial Tracking Attacks

**Jianlang Chen**[1]    **Xuhong Ren**[2]    **Qing Guo**[3†]    **Felix Juefei-Xu**[4§]    **Di Lin**[5]
**Wei Feng**[5]    **Lei Ma**[6,7]    **Jianjun Zhao**[1]
[1] Kyushu University, Japan    [2] Tianjin University of Technology, China
[3] CFAR and IHPC, Agency for Science, Technology and Research (A*STAR), Singapore
[4] GenAI, Meta, USA    [5] Tianjin University, China    [6] The University of Tokyo, Japan
[7] University of Alberta, Canada

## ABSTRACT

Visual object tracking plays a critical role in visual-based autonomous systems, as it aims to estimate the position and size of the object of interest within a live video. Despite significant progress made in this field, state-of-the-art (SOTA) trackers often fail when faced with adversarial perturbations in the incoming frames. This can lead to significant robustness and security issues when these trackers are deployed in the real world. To achieve high accuracy on both clean and adversarial data, we propose building a spatial-temporal implicit representation using the semantic text guidance of the object of interest extracted from the language-image model (*i.e.*, CLIP). This novel representation enables us to reconstruct incoming frames to maintain semantics and appearance consistent with the object of interest and its clean counterparts. As a result, our proposed method successfully defends against different SOTA adversarial tracking attacks while maintaining high accuracy on clean data. In particular, our method significantly increases tracking accuracy under adversarial attacks with around 90% relative improvement on UAV123, which is close to the accuracy on clean data. We have built a benchmark and released our code in `https://github.com/tsingqguo/robustOT`.

## 1 INTRODUCTION

Visual object tracking is a crucial technique in the field of vision intelligence, predicting the position and size of targeted objects in real-time video. It has found applications in various autonomous systems, including self-driving cars, unmanned aircraft, and robotics Over the years, significant advancements have been made in visual object tracking. State-of-the-art tracking methods now achieve high accuracy on challenging datasets by utilizing fully trained deep neural networks (DNNs). However, similar to the vulnerability of DNNs in image classification (Goodfellow et al., 2014; Carlini & Wagner, 2017; Guo et al., 2020a), deep tracking methods also face similar challenges (Wiyatno & Xu, 2019; Jia et al., 2021; Yan et al., 2020; Liang et al., 2020; Yin et al., 2022). Adversarial attacks can exploit this vulnerability by adding imperceptible perturbations to incoming frames, leading to incorrect predictions of the object's position by the deployed trackers. Such attacks pose security risks when deep trackers are integrated into automatic systems. These attacks could cause security issues when we embed deep trackers into the automatic systems. Therefore, it is crucial to enhance the robustness of deep trackers against adversarial tracking attacks.

There are two primary approaches to enhancing adversarial robustness in the context of image classification tasks. These include adversarial training (Kurakin et al., 2016; Tramèr et al., 2017; Rebuffi et al., 2021) and image preprocessing (Yuan & He, 2020; Nie et al., 2022; Ho & Vasconcelos, 2022). However, directly applying these methods to defend against adversarial tracking attacks is

---

[†]Qing Guo is the corresponding author (tsingqguo@ieee.org)
[§]Work done prior to joining Meta.

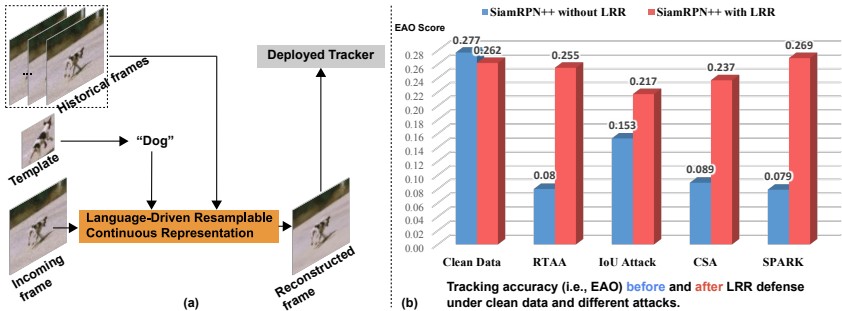

Figure 1: (a) shows the main idea of this work: we propose the language-driven resamplable continuous representation (LRR) that takes the template's text term and historical frames as inputs to reconstruct the incoming frame. (b) shows the results on VOT2019 (Kristan et al., 2019) with and without LRR under clean data and different attacks.

not straightforward. Adversarial training involves retraining deep models using a min-max optimization strategy, where the DNNs are exposed to more adversarial examples during the training process. However, this approach has certain limitations, such as a potential sacrifice in accuracy on clean data and increased time costs for training. Existing image preprocessing methods neglect the video sequence's temporal and the object template's semantic information, inadequately addressing the challenges of adversarial tracking attacks.

In this study, our focus is on a preprocessing-based solution to defend against tracking attacks. Specifically, we reconstruct the incoming frames and provide them to the deployed trackers to enhance adversarial robustness (See Figure 1 (a)). We argue that an effective preprocessing defense against tracking attacks should fulfill two criteria: (1) it should fully leverage the spatial and temporal contexts, which offer complementary appearance information, and (2) it should maintain semantic consistency with the object of interest as indicated by the initial frame, known as the object template. To achieve these objectives, we propose an approach based on the implicit representation (Chen et al., 2021), which effectively models the appearance of pixels based on their neighboring pixels. While existing implicit representation methods have shown promising results in image restoration, we propose a novel *language-driven resamplable continuous representation (LRR)* consisting of two key modules. First, we introduce the spatial-temporal implicit representation (STIR), enabling the reconstruction of any pixel at continuous spatial and temporal coordinates. This capability allows for the effective removal of adversarial perturbations and the achievement of appearance consistency with clean frames. Second, we propose a language-driven resample network (LResampleNet) that leverages the STIR. This network generates a new frame by feeding resampled continuous coordinates to the STIR, guided by the text from the object template. By aligning the resampled frame with the semantic information provided by the object template, we achieve semantic consistency. We conducted extensive experiments on three public datasets, demonstrating that our method significantly enhances the adversarial robustness of object trackers against four state-of-the-art adversarial attacks. Moreover, our approach maintains high accuracy on clean data, with the adversarial accuracy even matching or surpassing the clean accuracy. For instance, in the VOT 2019 results shown in Figure 1 (b), SiamRPN++ with LRR achieves an EAO of 0.283 under the SPARK attack, outperforming the 0.079 EAO achieved by SiamRPN++ without LRR and even surpassing the results on clean data.

## 2 BACKGROUND AND RELATED WORKS

**Visual object tracking.** Siamese trackers have become the current trend in visual object tracking tasks since they strike a great balance between tracking accuracy and efficiency (Li et al., 2018; Zhang & Peng, 2019; Fu et al., 2021; Cao et al., 2021). The SiamRPN (Li et al., 2018) algorithm approaches VOT as a one-shot detection problem and was the first to introduce a region proposal network (RPN (Ren et al., 2015)) into the tracking arena. By incorporating RPN, SiamRPN mitigates the need for heavy multi-scale correlation operations, resulting in high-speed and accurate tracking performance. SiamRPN+ (Zhang & Peng, 2019) and SiamRPN++ (Li et al., 2019) propose the incorporation of a cropping residual unit and a spatial-aware sampling strategy, enabling the Siamese RPN framework to benefit from modern backbones and significantly enhance the performance of

the Siamese tracker. In this work, we evaluate the effectiveness of our defense mechanism on two trackers from the SiamRPN++ family that are popular within adversarial research. In recent years, transformer-based trackers (Ye et al., 2022; Lin et al., 2022; Cui et al., 2022; Mayer et al., 2022) have demonstrated remarkable tracking accuracy. Our initial results indicate that our method remains effective for transformer-based trackers.

**Adversarial tracking attacks.** In recent years, the broad applications of visual object tracking have prompted a wide range of studies on the robustness of visual object trackers (Wiyatno & Xu, 2019; Guo et al., 2019). AD2Atk (Fu et al., 2022) focuses on generating adversarial examples during the resampling of the search path image. EfficientAdv (Liang et al., 2020) presents an end-to-end network that employs a novel drift loss in conjunction with the embedded feature loss to attack the tracker. DIMBA (Yin et al., 2022) proposes a black-box attack that uses reinforcement learning to localize crucial frames accurately. CSA (Yan et al., 2020) employs a well-crafted cooling-shrinking loss to train an efficient adversarial perturbation generator. RTAA (Jia et al., 2020) conducts a frame-by-frame attack, introducing temporal perturbation into the original video sequences and significantly reducing the tracking performance. SPARK (Guo et al., 2020b) is designed to attack online trackers by imposing a $L_p$ constraint on perturbations while calculating them incrementally based on previous attacks. IoU (Jia et al., 2021) creates perturbations by utilizing temporally correlated information and incrementally adding noise from the initial frame to subsequent frames.

These advanced attackers exploit the unique characteristics of VOT, thereby making defense methods, originally adapted from the image classification domain, difficult to apply effectively. In response to this, our approach seeks to use spatial-temporal representation to leverage the information concealed within inter-frame relationships.

**Adversarial robustness enhancement** Approaches for enhancing robustness typically fall into two main categories: adversarial training and input preprocessing during testing time. The adversarial training approach introduces adversarial perturbations during training (Goodfellow et al., 2014; Kurakin et al., 2016; Madry et al., 2017; Tramèr et al., 2017; Athalye et al., 2018; Rebuffi et al., 2021), which are usually computationally expensive. The input preprocessing methods (Yuan & He, 2020; Nie et al., 2022; Ho & Vasconcelos, 2022) are employed to remove the adversarial perturbations, and thus enhance the robustness. However, these methods are mainly designed for image classification tasks and cannot be used to defend against adversarial tracking attacks directly. For example, DiffPure (Nie et al., 2022) utilizes diffusion models for adversarial purification. While it exhibits promising results in image classification tasks, its intensive computational demands make it infeasible for video tasks. The purification process for a single image of $256 \times 256$ pixels requires approximately 26 seconds, which equates to a processing speed of 0.04 fps for video frame processing. We provide an empirical study in A.7 by using the DiffPure for tracking defense. Unlike previous enhancement approaches, our method leverages historical information from the object template to build a novel defense pipeline against video-specific adversarial attacks.

**Implicit representation.** Implicit representation has been extensively employed in the modeling of 3D object shapes and structures, where a 3D object is typically represented by a multilayer perceptron (MLP) that maps coordinates to signals. Inspired by its success in 3D tasks, recent studies have proposed the application of implicit representation in 2D tasks. (Chen et al., 2021) proposed the Local Implicit Image Function (LIIF), which generates a continuous representation for super-resolution. Lee & Jin (2022) improves LIIF by adding high-frequency information in Fourier space. (Ho & Vasconcelos, 2022) emerged with an adversarial defense method that eliminates adversarial perturbations by utilizing local implicit functions. Both DISCO and LIIF perform their tasks based on local implicit image representation. Contrastingly, our work proposes a novel approach that extends local implicit representation into spatial-temporal implicit representation.

## 3 LANGUAGE-DRIVEN RESAMPLABLE CONTINUOUS REPRESENTATION

### 3.1 OVERVIEW

Given a live video, an object tracker aims to predict the position and size of the object of interest, which is indicated by an object template **T** cropped from the first frame. Adversarial tracking attacks usually inject adversarial perturbations into incoming frames, leading to incorrect tracking results. In this section, we propose the *language-driven resamplable continuous representation (LRR)* against

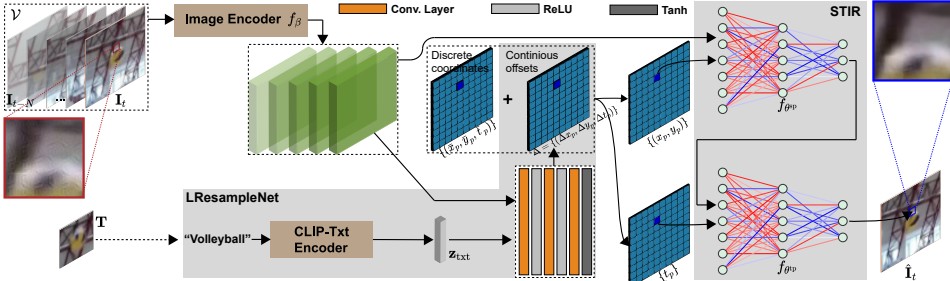

Figure 2: Pipeline of proposed language-driven resamplable continuous representation (LRR) that contains two key parts, *i.e.*, spatial-temporal implicit representation (STIR) and language-driven ResampleNet (LResampleNet). STIR takes continuous spatial and temporal coordinates as inputs (See point center at the blue rectangle) and estimates the corresponding color value.

adversarial tracking attacks. The intuitive idea is that we try to reconstruct an incoming frame to remove the penitential adversarial perturbations while maintaining its semantic consistency to the object template indicated in the first frame. Given an incoming frame $\mathbf{I}_t$ that may be corrupted by adversarial perturbation, we try to reconstruct it and get a new counterpart $\hat{\mathbf{I}}_t$. The objective contains two components: the first one is to remove the adversarial perturbations and encourage the reconstructed frame to have the same appearance as its clean counterpart. The second component is to make the semantic information of the reconstructed frame and the object template be consistent.

We have the following challenges when addressing the two objectives: *First*, as we are handling a live video, the historical frames should provide complementary spatial and temporal information, that is, a perturbed pixel usually has a similar appearance to its spatial and temporal neighboring pixels that can be used to reconstruct the perturbed pixels. *The key problem is how to build a bridge between the spatial & temporal axes and pixel appearances,* which should have a high generalization to adapt to different pixel intensities or colors. *Second*, in terms of semantic consistency, a straightforward solution is to extract the deep features (*e.g.*, VGG features) of the incoming frame and the object template, respectively, and then encourage the two features to be similar. However, such a solution could only approach deep feature consistency instead of semantic consistency. There are two reasons preventing this solution: (1) the deep features are not exactly aligned with the semantic space. (2) the deep features themselves are still vulnerable to adversarial perturbations.

To address the first challenge, we propose to build a spatial-temporal implicit representation (See STIR in Section 3.2) that enables the reconstruction of any pixels at continuous spatial and temporal coordinates, which can remove the adversarial perturbations effectively and achieve appearance consistency to the clean counterpart (Chen et al., 2021; Ho & Vasconcelos, 2022). Regarding the second challenge, we propose a language-driven resample network (*i.e.*, LResampleNet in Section 3.3) based on the built spatial-temporal implicit representation, which is able to generate a new frame by feeding resampled continuous coordinates to the STIR under the guidance of the text from the object template. Such a module makes the resampled frame have the same semantic text as the object template, naturally leading to semantic consistency. We display the whole pipeline in Figure 2.

## 3.2 SPATIAL-TEMPORAL IMPLICIT REPRESENTATION (STIR)

Given an image sequence $\mathcal{V} = \{\mathbf{I}_\tau \in \mathbb{R}^{H \times W}\}_{\tau=t-N}^t$ containing the $t$th frame and its historical $N$ neighboring frames, we aim to construct an implicit representation for the sequence, *i.e.*, $\hat{\mathcal{V}}$, which maps the spatial and temporal coordinates of a pixel (*i.e.*, $\mathbf{p} = (x_p, y_p, \tau_p)$) in the continuous domain to the corresponding RGB value (*i.e.*, $\hat{\mathcal{V}}(\mathbf{p})$). To this end, we propose to extend the recent local implicit image representation (Chen et al., 2021; Ho & Vasconcelos, 2022) to the spatial-temporal domain. In a straightforward way, we can formulate the task as

$$\hat{\mathcal{V}}(\mathbf{p}) = \sum_{\mathbf{q} \in \mathcal{N}_\mathbf{p}} \omega_\mathbf{q} f_\theta(\mathbf{z}_\mathbf{q}, \text{dist}(\mathbf{p}, \mathbf{q})), \tag{1}$$

where $\mathbf{p} = (x_p, y_p, \tau_p)$ is the coordinates of a pixel in the continuous spatial and temporal domain, that is, $x_p \in [0, H-1]$ and $y_p \in [0, W-1]$ can be non-integer and determines the spatial position of the pixel while $\tau_p \in [t-N, t]$ can be non-integer and decide its temporal location. The set $\mathcal{N}_\mathbf{p}$

contains neighboring pixels of the pixel $\mathbf{p}$ in $\mathcal{V}$. The vector $\mathbf{z_q}$ denotes the feature of the pixel $\mathbf{q}$, and the function $\text{dist}(\mathbf{p}, \mathbf{q})$ measures the spatial distance between the two pixels (*e.g.*, Euclidean distance). The function $f_\theta$ is parameterized as an MLP. Intuitively, the function $f_\theta(\mathbf{z_q}, \text{dist}(\mathbf{p}, \mathbf{q}))$ is to map the feature of neighboring pixel $\mathbf{q}$ to the color of $\mathbf{p}$ based on their spatial distance. All generated color values are weightedly aggregated and the weight $\omega_\mathbf{q}$ is determined by the volume ratio of the cube formed by $\mathbf{p}$ and $\mathbf{q}$ to the total neighboring volume.

The complexity of the above formulation (*e.g.*, Equation 1) is directly related to the size of the neighboring set. For example, we consider $K \times K$ spatial neighboring pixels across the $N$ neighboring temporal frames. Then, the complexity of a pixel's reconstruction is $\mathcal{O}(NK^2)$. To alleviate the computing costs, we propose to decompose the reconstruction along the spatial and temporal domains and reformulate the Equation 1. Specifically, we first build a spatial implicit representation that estimates the color values of a spatial location across all neighboring frames; that is, we have

$$\hat{\mathcal{V}}(\mathbf{p}_{(t-N:t)}) = \sum_{(x_q,y_q)\in\mathcal{N}_{(x_p,y_p)}} \omega^{\text{sp}}_{(x_q,y_q)} f_{\theta^{\text{sp}}}(\mathbf{z}_{\mathbf{q}_{(t-N:t)}}, \text{dist}((x_p,y_p),(x_q,y_q))), \tag{2}$$

where $\mathbf{p}_{(t-N:t)} = [(x_p, y_p, t-N), \ldots, (x_p, y_p, t)]$ and $\hat{\mathcal{V}}(\mathbf{p}_{(t-N:t)})$ preserves the $N$ color values of pixels at poistion $(x_p, y_p)$ across all temporal frames. The term $\mathbf{z}_{\mathbf{q}_{(t-N:t)}}$ concatenates the features of all pixels at location $(x_p, y_p)$ across all temporal frames. The function $f_{\theta^{\text{sp}}}$ is an MLP with the parameter $\theta^{\text{sp}}$, and the weight $\omega^{\text{sp}}_{(x_q,y_q)}$ is determined by the area ratio of the rectangle formed by $(x_p, y_p)$ and $(x_q, y_q)$ to the total neighboring areas as done in (Chen et al., 2021). After getting $\hat{\mathcal{V}}(\mathbf{p}_{(t-N:t)})$, we further build a temporal implicit representation that can estimate the color value of the pixel $\mathbf{p} = (x_p, y_p, \tau_p)$, that is, we have

$$\hat{\mathcal{V}}(\mathbf{p}) = \sum_{\tau_q \in [t-N,t]} \omega^{\text{tp}}_{\tau_q} f_{\theta^{\text{tp}}}(\hat{\mathcal{V}}(\mathbf{p}(t - N : t))[\tau_q], \text{dist}(\tau_p, \tau_q)), \tag{3}$$

where $\mathcal{V}(\mathbf{p}(t - N : t))[\tau_q]$ is the $\tau_q$th element in $\mathcal{V}(\mathbf{p}(t - N : t))$, and $f_{\theta^{\text{tp}}}(\cdot)$ is also an MLP to map the predicted $\mathcal{V}(\mathbf{p}(t - N : t))[\tau_q]$ to color value of the pixel $\mathbf{p}$. Compared with Equation 1, the complexity of Equation 2 and Equation 3 is reduced to $\mathcal{O}(K^2 + N)$.

We can simplify the developed STIR (*i.e.*, Equation 2 and Equation 3) as

$$\hat{\mathcal{V}}(\mathbf{p}) = \text{STIR}(\mathbf{p}, \mathcal{V}|f_\beta, f_{\theta^{\text{sp}}}, f_{\theta^{\text{tp}}}) \tag{4}$$

where $f_\beta$ is an encoder network to extract pixels' features (*i.e.*, $z_q$ in Equation 1). Once we train the parameters of $f_\beta$, $f_{\theta^{\text{sp}}}$, $f_{\theta^{\text{tp}}}$, we can generalize STIR to build implicit representations for arbitrary image sequences.

### 3.3 LANGUAGE-DRIVEN RESAMPLENET (LRESAMPLENET)

With the STIR, we can resample the $t$th frame by

$$\hat{\mathbf{I}}_t(\bar{\mathbf{p}}) = \text{STIR}(\mathbf{p}, \mathcal{V}|f_\beta, f_{\theta^{\text{sp}}}, f_{\theta^{\text{tp}}}), \text{with } \mathbf{p} = \bar{\mathbf{p}} + \Delta\mathbf{p} \tag{5}$$

where $\bar{\mathbf{p}} = (\bar{x}_p, \bar{y}_p, t)$ is the discrete coordinate of the $t$th frame, that is, $\bar{x}_p$, and $\bar{y}_p$ are integers sampled from $[0, H - 1]$ and $[0, W - 1]$, respectively. Note that, we fix the temporal coordinate as $t$ since we handle the $t$th frame. $\Delta\mathbf{p}$ are continuous offsets to generate continuous coordinates (*i.e.*, $\mathbf{p}$) based on the integer coordinates $\bar{\mathbf{p}}$. Hence, if we iterate through all discrete coordinates within the frame $\mathbf{I}_t$, we can reconstruct the $t$th frame and get $\hat{\mathbf{I}}_t$. The key problem is how to predict the offset $\Delta\mathbf{p}$. We propose to use the language extracted from the template $\mathbf{T}$ and the pixel's feature to guide the resampling, that is, to generate the offset for each pixel in $\mathbf{I}_t$.

Specifically, we initiate the process by employing CLIP (Radford et al., 2021)'s image encoder to extract the template (*i.e.*, $\mathbf{T}$)'s embedding. Subsequently, given a set of texts encompassing potential categories of the object, we compare the template's embedding with the embeddings of all the texts. Following this, we select the text embedding that exhibits the highest similarity with the template's embedding as the $\mathbf{z}_{\text{txt}}$. Note that the text set can be updated based on different application scenarios, and alternative vision-language models or image caption methods can also be employed to achieve

the same objective. After that, we design a convolutional neural network denoted as *language-driven resampleNet (LResampleNet)* that takes the template's text embedding and pixel's feature embedding as inputs and predicts the offset; that is, we have

$$\Delta = \text{LResampleNet}(\mathbf{Z}, \mathbf{z}_{\text{txt}}) \tag{6}$$

where $\mathbf{Z} \in \mathbb{R}^{H \times W \times C}$ contains the $C$-channel features of $HW$ pixels in $\mathbf{I}_t$ and is extracted via the encoder network $f_\beta(\cdot)$, and $\mathbf{z}_{\text{txt}} \in \mathbb{R}^{1 \times M}$ is the text embedding of the object template. In practice, we concatenate each pixel's feature with the text embedding and feed them to the LResampleNet. The output $\Delta \in \mathbb{R}^{H \times W \times 3}$ contains the offsets of all pixels.

### 3.4 IMPLEMENTATION DETAILS

**Architectures.** We set the $f_{\theta^{\text{sp}}}$ and $f_{\theta^{\text{tp}}}$ are five-layer MLPs with a ReLU activation layer and the hidden dimensions are 256. We use the network of (Lim et al., 2017) without the upsampling modules as the encoder for extracting pixel features (*i.e.*, $f_\beta$), which can generate a feature with the same size as the input image.

**Loss function.** Given an attacked image sequence $\mathcal{V} = \{\mathbf{I}_\tau\}_{\tau = t-N}^t$ and the object template $\mathbf{T}$, we obtain the reconstructed $t$th frame $\hat{\mathbf{I}}_t$. When we have the clean version of $\hat{\mathbf{I}}_t$ (*i.e.*, $\mathbf{I}_t^*$), we follow existing works and only use the $L_1$ loss function to train the STIR and LResampleNet. Intuitively, following the objectives in Section 3.1, we can add a consistency loss for the features of $\hat{\mathbf{I}}_t$ and $\mathbf{T}$ but we do not see clear benefits.

**Training datasets.** We employ three widely-used datasets, *i.e.*, ImageNet-DET (Russakovsky et al., 2015), ImageNet-VID, and YouTube-BoundingBoxes (Real et al., 2017) to train the STIR. Specifically, given a randomly sampled video, we randomly select five continuous frames in the video to form an image sequence and crop the object template $\mathcal{T}$ from another randomly chosen frame. Then, we add adversarial perturbations to the image sequence and regard the perturbed sequence as the $\mathcal{V}$ in Equation 4. Here, we apply the FGSM attack on a pre-trained SiamRPN++ with ResNet50 tracker to produce adversarial perturbations. After that, we have a pair of $\mathcal{V}$ and $\mathcal{T}$ as the training sample. We have sampled around 490,000 pairs for training STIR and LResampleNet, and 20,000 pairs as the validation set. We train the STIR and LResampleNet independently since they have different functionalities, and joint training could hardly get good results for both modules. Besides, ImageNet-DET is an image dataset and we perform random translations on its images to get an image sequence to enlarge the training datasets.

**Other details.** We train and perform our method on a server with an NVIDIA RTX A6000 GPU and an Intel Core i9-10980XE 3.0GHz CPU using Pytorch (Paszke et al., 2019). In alignment with the tracker's design, we have configured the reconstruction range to be the search region rather than the entire image, resulting in a significant reduction in time costs.

**LRR for adversarial tracking defense.** LRR has a high generalization. After training LRR, we can use it to defend against diverse attacks for different trackers on any tracking dataset. Specifically, given an incoming frame, we can employ the Equation 5 and Equation 6 to reconstruct it and feed it to subsequent trackers to estimate the object's location and size.

## 4 EXPERIMENTAL RESULTS

We conduct a series of experiments to evaluate LRR's defensive efficacy under various previously discussed settings, reporting the average results from three independent trials.

**Testing datasets.** For evaluate the effectiveness of adversarial defense approach, we utilized three widely used tracking datasets: OTB100 (Wu et al., 2015), VOT2019 (Kristan et al., 2019), and UAV123 (Mueller et al., 2016). VOT2019 and OTB100 are popular tracking datasets that consist of 60 and 100 videos, respectively. UAV123 dataset focuses on object tracking in videos captured by uncrewed aerial vehicle cameras, containing 123 videos.

**Trackers and attacks.** Given the variance in adversarial attacks on VOT tasks across both algorithms and implementations, it is crucial to employ representative trackers to facilitate a comprehensive and impartial assessment of adversarial attack resilience. This approach also serves to demonstrate the general efficacy of our proposed defense mechanism. To this end, we select trackers from the SiamRPN++ family: SiamRPN++ with ResNet50 and SiamRPN++ with MobileNetV2, and identify four challenging attackers, the IoU Attack (Jia et al., 2021), SPARK (Guo et al., 2020b),

Table 1: Comparing LRR with baselines on OTB100, VOT2019, and UAV123 under Four Attacks.

| SiamRPN++ | Defenses | OTB100 Prec. (%) | | | | | VOT2019 EAO | | | | | UAV123 Prec. (%) | | | | |
|---|---|---|---|---|---|---|---|---|---|---|---|---|---|---|---|---|
| | | Cln. | RTAA | IoU | CSA | SPARK | Cln. | RTAA | IoU | CSA | SPARK | Cln. | RTAA | IoU | CSA | SPARK |
| Res50 | wo.Def | **91.4** | 32.7 | 75.9 | 47.2 | 69.8 | **0.277** | 0.080 | 0.153 | 0.089 | 0.079 | **79.5** | 41.2 | 70.5 | 46.5 | 40.8 |
| | AT$_{FGSM}$ | 85.1 | 53.5 | 77.6 | 60.7 | 69.6 | 0.214 | 0.100 | 0.176 | 0.109 | 0.078 | 77.4 | 48.9 | 72.5 | 44.9 | 41.3 |
| | AT$_{PGD}$ | 81.8 | 50.2 | 76.8 | 62.3 | 61.0 | 0.218 | 0.090 | 0.171 | 0.125 | 0.057 | 79.4 | 55.6 | 74.0 | 68.8 | 37.0 |
| | AT$_{CSA}$ | 84.3 | 52.2 | 77.2 | 80.9 | 65.1 | 0.251 | 0.090 | 0.164 | 0.152 | 0.072 | 76.8 | 43.2 | 70.6 | 74.2 | 34.5 |
| | DISCO | 86.0 | 86.3 | 78.6 | 83.6 | 85.7 | 0.249 | 0.244 | 0.190 | 0.204 | 0.248 | 79.1 | 76.8 | 75.9 | 77.7 | 76.0 |
| | LRR | 87.8 | **86.9** | **85.3** | **89.4** | **89.3** | 0.262 | **0.255** | **0.217** | **0.237** | **0.269** | 79.3 | **77.7** | **78.6** | **81.8** | **79.3** |
| MobileV2 | wo.Def | 85.5 | 25.6 | 67.8 | 40.9 | 32.2 | **0.267** | 0.062 | 0.125 | 0.083 | 0.037 | **80.3** | 39.3 | 66.2 | 42.2 | 22.5 |
| | AT$_{FGSM}$ | 79.4 | 35.6 | 72.6 | 63.1 | 30.1 | 0.213 | 0.070 | 0.144 | 0.121 | 0.041 | 78.4 | 39.3 | 67.1 | 64.1 | 21.6 |
| | AT$_{PGD}$ | 77.7 | 34.9 | 72.0 | 73.0 | 23.9 | 0.202 | 0.075 | 0.122 | 0.130 | 0.035 | 77.9 | 36.7 | 67.7 | 71.3 | 18.7 |
| | AT$_{CSA}$ | 79.4 | 34.2 | 60.2 | 76.6 | 35.7 | 0.263 | 0.078 | 0.097 | 0.137 | 0.037 | 74.8 | 43.4 | 56.4 | 70.9 | 14.8 |
| | DISCO | 82.9 | 78.7 | 75.0 | 79.9 | 80.1 | 0.175 | 0.161 | 0.132 | 0.166 | 0.208 | 74.6 | 76.2 | 73.3 | 72.9 | 75.3 |
| | LRR | 85.6 | 83.2 | 82.1 | 83.9 | 85.4 | 0.240 | **0.205** | **0.166** | **0.223** | **0.239** | 79.1 | **78.7** | **75.9** | **76.2** | **79.0** |

| | Ground Truth | | Results before Defense | | Results after Defense |
|---|---|---|---|---|---|

Figure 3: Visualization comparison before & after LRR defense for SiamRPN++ under CSA attack.

CSA (Yan et al., 2020), and RTAA (Jia et al., 2020), which are known to deliver robust performance against SiamRPN++ trackers. We detail the implementations of these attacks in A.9.

**Defence baselines.** To assess the effectiveness of our proposed method comprehensively, we compare it against adversarial fine-tuning techniques and SOTA adversarial defense approach. Adversarial fine-tuning, as outlined by (Goodfellow et al., 2014), is a strategy that trains a model with both clean and adversarial examples, thereby enhancing the model's resilience against attacks. For the adversarial fine-tuning baseline, we employ FGSM (Goodfellow et al., 2014), PGD (Madry et al., 2017), and CSA (Yan et al., 2020) to generate adversarial examples and augment the training data, thereby enabling the model to fortify its defenses against adversarial attacks. Both PGD and FGSM add minimal calculated perturbation to the input image based on the gradient of the tracker model's loss concerning the input, while CSA uses its perturbation generator to inject adversarial examples, progressively reducing the confidence of the tracker's backbone. For the adversarial defense method, we adapt the SOTA method, DISCO (Ho & Vasconcelos, 2022), for tracking tasks, using it to predict each pixel's RGB value through local implicit functions, thus defending against attacks. We incorporate DISCO as a frame processor into our adversarial tracking attacks defense task.

## 4.1 COMPARISON RESULTS

LRR achieves SOTA performance over the baselines, as detailed in Table 1, which analyzes adversarial defense under four attacks across three datasets and two SiamRPN++ family trackers. The LRR setup follows the approach in Section 3.4. The table illustrates that SiamRPN++ trackers can be compromised, impacting precision on OTB100 and UAV123 and Expected Average Overlap (EAO) on VOT2019. FGSM and PGD, as adversarial fine-tuning approaches, provide minimal defense, decreasing performance even on non-attacked inputs. While CSA fine-tuning improves defense against its generator's examples, it underperforms under other attacks. Overall, the adversarial fine-tuning baselines present a marginally successful defense against IoU and CSA but are ineffective against RTAA and SPARK. Meanwhile, DISCO displays robust defense against all attack types but is outperformed by LRR due to its inability to leverage information between frames. To validate the effectiveness further, we compare the visualizations of DISCO and LRR at both the image level and the response map level in the supplementary material A.3. The results demonstrate that LRR can achieve higher consistency at the semantic and image quality levels than DISCO.

Table 2: Comparison between STIR alone and LResampleNet on VOT2019, OTB100, and UAV123.

| SiamRPN++ | Attacks | OTB100 Prec. (%) | | | VOT2019 EAO | | | UAV123 Prec. (%) | | |
|---|---|---|---|---|---|---|---|---|---|---|
| | | Org. | wo.LResample | LResample | Org. | wo.LResample | LResample | Org. | wo.LResample | LResample |
| Res50 | wo.Atk | 91.4 | **88.1** | 87.8 | 0.277 | **0.268** | 0.262 | 79.5 | **79.5** | 79.3 |
| | RTAA | 32.7 | 85.9 | **86.9** | 0.080 | 0.247 | **0.255** | 41.2 | 77.0 | **77.7** |
| | IoUAttack | 75.9 | 84.0 | **85.3** | 0.153 | 0.213 | **0.217** | 70.5 | 78.2 | **78.6** |
| | CSA | 47.2 | 85.9 | **89.4** | 0.089 | 0.219 | **0.237** | 46.5 | 79.7 | **81.8** |
| | SPARK | 69.8 | 87.7 | **89.3** | 0.079 | 0.256 | **0.269** | 40.8 | 77.9 | **79.3** |
| MobileV2 | wo.Atk | 85.5 | 85.5 | **85.6** | 0.267 | 0.238 | **0.240** | 80.3 | 78.4 | **79.1** |
| | RTAA | 25.6 | 80.7 | **83.2** | 0.062 | 0.204 | **0.205** | 39.3 | 77.7 | **78.7** |
| | IoUAttack | 67.8 | 79.0 | **82.1** | 0.125 | 0.163 | **0.166** | 66.2 | 74.9 | **75.9** |
| | CSA | 40.9 | 79.8 | **83.9** | 0.083 | 0.216 | **0.223** | 42.2 | 75.7 | **76.2** |
| | SPARK | 32.2 | 83.7 | **85.4** | 0.037 | 0.225 | **0.239** | 22.5 | 78.2 | **79.0** |

Table 3: Comparison of ResampleNet with & without language guidance on three datasets.

| SiamRPN++ | Attacks | OTB100 Prec. (%) | | | VOT2019 EAO | | | UAV123 Prec. (%) | | |
|---|---|---|---|---|---|---|---|---|---|---|
| | | Org. | wo.Lang. | Lang. | Org. | wo.Lang. | Lang. | Org. | wo.Lang. | Lang. |
| Res50 | wo.Atk | 91.4 | **88.1** | 87.8 | 0.277 | **0.270** | 0.262 | 79.5 | **79.6** | 79.3 |
| | RTAA | 32.7 | 86.1 | **86.9** | 0.080 | 0.250 | **0.255** | 41.2 | 77.4 | **77.7** |
| | IoUAttack | 75.9 | 84.4 | **85.3** | 0.153 | 0.219 | **0.217** | 70.5 | 78.4 | **78.6** |
| | CSA | 47.2 | 86.0 | **89.4** | 0.089 | 0.222 | **0.237** | 46.5 | 79.8 | **81.8** |
| | SPARK | 69.8 | 87.8 | **89.3** | 0.079 | 0.254 | **0.269** | 40.8 | 78.1 | **79.3** |
| MobileV2 | wo.Atk | 85.5 | **86.1** | 85.6 | 0.267 | 0.238 | **0.240** | 80.3 | **79.6** | 79.1 |
| | RTAA | 25.6 | 81.2 | **83.2** | 0.062 | 0.201 | **0.205** | 39.3 | 78.0 | **78.7** |
| | IoUAttack | 67.8 | 79.0 | **82.1** | 0.125 | 0.151 | **0.166** | 66.2 | 75.3 | **75.9** |
| | CSA | 40.9 | 81.1 | **83.9** | 0.083 | 0.219 | **0.223** | 42.2 | 76.0 | **76.2** |
| | SPARK | 32.2 | 84.5 | **85.4** | 0.037 | 0.228 | **0.239** | 22.5 | 78.4 | **79.0** |

## 4.2 ABLATION STUDY AND DISCUSSION

In this section, we explore the effectiveness of the components in our LRR (Language-Driven Re-sampling Network), specifically discussing the individual contributions of the resampling network, language-driven approach, and spatial-temporal information toward the defense mechanism.

**Overall results.** LRR has demonstrated robust defense against adversarial attacks. Employing the VOT2019's Expected Average Overlap (EAO) metric, a composite measure of Accuracy and Robustness, it is evident from Table 1 that our defenses significantly enhanced EAO. Following the implementation of our defense, the average EAO value under attack increased to 89% and 81% for the SiamRPN++ with ResNet50 and MobileNetV2 trackers, respectively. Additionally, using precision as a metric for the OTB100 and UAV123 datasets, our defense approach has shown a boost in precision to 90% across all attackers and trackers, highlighting its effectiveness. Furthermore, we extend our evaluation to four additional widely used datasets, including one large-scale dataset, as detailed in A.1. This extended evaluation demonstrates the effective transferability of our method across diverse datasets. In A.8, we also compare with image resizing and compression-based defense methods, which further demonstrates the advantages of our method.

**Illustrative Overview of Defense Results.** Figure 3 qualitatively demonstrates the defense results achieved by LRR. Our method removes the adversarial textures effectively and makes the tracker localize the object of interest accurately. In A.3, we delve further into visualizations on correlation maps and discuss in greater depth the impact of our method on adversarial defense.

**Effectiveness of resampling.** To validate the effectiveness of our primary contribution, we conducted experiments to demonstrate the influence of the LResampleNet in LRR. Given STIR's independent training with LResampleNet, it should estimate perturbations utilizing spatial-temporal information. We evaluated STIR without resampling, following the experimental settings of previous evaluations. In Table 2, we present the increase in precision for the OTB100 and UAV123 datasets and the rise in EAO value for VOT2019. In Table 2, results indicate that tracking outcomes without the LResampleNet are less effective than LRR in defending against adversarial tracking attacks. A more detailed discussion on this is articulated in A.2.

**Effectiveness of language guidance.** When introducing the resampling mechanism into our pipeline in Section 3.4, we used language to establish a connection between incoming frames and the tracking template, constituting a major contribution to our work. Since we feed both pixel embedding and text embedding to the resampling network, we aim to validate the effectiveness of our language-driven approach. We designed a resampling network without text embedding (ResampleNet), allowing pixel embedding to serve as the sole input, replacing the LResampleNet in our existing pipeline. As shown in Table 3, the use of ResampleNet guidance appears to be less effective when compared to our LRR pipeline. However, compared to the pipeline that uses STIR alone, ResampleNet demonstrates an enhanced defense against adversarial tracking attacks. The primary reason for this is ResampleNet's ability to estimate adversarial perturbations by leveraging

Table 4: Comparison of STIR with different settings of $N$, evaluated on OTB100 and VOT2019.

| SiamRPN++ | Attacks | OTB100 Prec. (%) | | | | | | VOT2019 EAO | | | | | |
|---|---|---|---|---|---|---|---|---|---|---|---|---|---|
| | | Org. | $N=1$ | $N=2$ | $N=3$ | $N=4$ | $N=5$ | Org. | $N=1$ | $N=2$ | $N=3$ | $N=4$ | $N=5$ |
| | wo.Atk | 91.4 | 86.5 | 87.3 | 87.1 | 87.3 | 88.1 | 0.277 | 0.237 | 0.251 | 0.255 | 0.267 | 0.268 |
| | RTAA | 32.7 | 84.9 | 85.0 | 85.6 | 85.6 | 85.9 | 0.080 | 0.240 | 0.241 | 0.241 | 0.245 | 0.247 |
| Res50 | IoUAttack | 75.9 | 78.7 | 79.2 | 80.9 | 83.1 | 84.0 | 0.153 | 0.190 | 0.191 | 0.208 | 0.211 | 0.213 |
| | CSA | 47.2 | 82.6 | 82.8 | 83.3 | 85.2 | 85.9 | 0.089 | 0.203 | 0.204 | 0.215 | 0.216 | 0.219 |
| | SPARK | 69.8 | 85.5 | 85.8 | 86.3 | 87.0 | 87.7 | 0.079 | 0.245 | 0.249 | 0.251 | 0.252 | 0.256 |

the implicit continuous representation from the input pixel embedding.

**Effectiveness of spatial-temporal information.** To validate STIR in learning spatial-temporal information, we trained it separately by altering the input frame length $N \in \{1, 2, 3, 4, 5\}$ from the training dataset described in Section 3.4. To assess the influence of LResampleNet, we evaluated these STIR models independently without the integration of our LRR on OTB100 and VOT2019 datasets using SiamRPN++ with ResNet50 tracker. The results presented in Table 4, reveal that as the number of frame inputs length $N$ increases, STIR demonstrates an enhanced capability to defend against adversarial tracking attacks. This suggests that STIR is able to extract more hidden information from spatial-temporal information brought by input frames, thereby serving a better purpose in constructing video frame RGB values from it.

**Transferability to transformer-based trackers.** To clarify the transferability of our LRR approach, we adapted our method to the recently proposed transformer-based tracker model, ToMP-50 (Mayer et al., 2022).

Specifically, we employed RTAA to attack ToMP-50 and applied our LRR method for defense, evaluating the results across three different datasets. The results, delineated in Table 4, underscore the transferability of our proposed method, sustaining its efficacy even when incorporated with newly developed tracking models. A detailed discussion can be found in A.4.

Table 5: Defense on ToMP-50 across 3 datasets

| ToMP | Attacks | OTB100 Prec. (%) | | VOT2019 EAO | | UAV123 Prec. (%) | |
|---|---|---|---|---|---|---|---|
| | | Org. | LRR | Org. | LRR | Org. | LRR |
| ToMP-50 | wo.Atk | 90.1 | **89.8** | 0.556 | **0.547** | 88.2 | **87.8** |
| | RTAA | 61.8 | **90.0** | 0.337 | **0.552** | 58.5 | **88.0** |

**Defense efficiency.** LRR addresses attacks via the elimination of perturbations at testing time. This strategy allows our method to be easily integrated into various existing tracking task pipelines, which also raises the concern of additional computational consumption. We report the time cost of our methods in Table 6. Using our proposed method as a standalone frame processor, our STIR defense can maintain processing at approximately 29 fps. In comparison, LRR operates at around 25 fps. This allows for the facilitation of online tracking adversarial defense capability. For a more detailed discussion, please refer to A.6.

## 5 CONCLUSION

In this work, we have developed a novel implicit representation, *i.e.*, language-driven resamplable continuous representation (LRR), against state-of-the-art adversarial tracking attacks. We first built a spatial-temporal implicit representation (STIR) to utilize the spatial-

Table 6: Average time costs on OTB100.

| SiamRPN++ | Track | Attack | | | | Defense | |
|---|---|---|---|---|---|---|---|
| | - | RTAA | IoUAttack | CSA | SPARK | STIR | LRR |
| Res50 | 16 | 215 | 1184 | 4 | 76 | 34 | 39 |
| MobileV2 | 13 | 118 | 667 | 4 | 62 | 34 | 39 |

temporal neighboring pixels for effective appearance reconstruction. Then, we designed the language-driven ResampleNet to encourage semantic consistency between the reconstructed frame and the object template. After training on large-scale datasets, our method can be used to defend against different attacks for different trackers on different testing datasets. Impressively, our method has successfully defended four state-of-the-art attacks and let the adversarial accuracy approach the clean accuracy while maintaining the high accuracy on the clean data.

**Limitations.** As an extra module, the proposed method inevitably increases the computing and time costs. In the future, we can explore approaches to decrease costs. Besides, the generalization to non-noise-based attacks like motion blur (Guo et al., 2021) should be future studied. Furthermore, in recent years, researchers have increasingly directed their attention toward natural language-specified visual object tracking (Wang et al., 2021), which offers greater flexibility in real-world scenarios. However, existing attack and defense methods predominantly focus on template-based trackers, overlooking this emerging trend. Future research endeavors should aim to bridge this gap.

## 6 REPRODUCIBILITY STATEMENT

To facilitate the reproducibility of our approach, we have open-sourced our code and provided a benchmark that includes our method, which is accessible via `https://github.com/tsingqguo/robustOT`. This repository contains the essential evaluation code, along with comprehensive instructions to facilitate the deployment of the proposed methods and the establishment of the evaluation environment. The repository also includes a pre-trained model, allowing for direct replication of the demonstrated results.

All implementation details are meticulously described in Section 3.4. The thorough documentation, along with the availability of the benchmark and pre-trained model, aims to assist in the validation and replication of the presented results.

## ACKNOWLEDGMENT

This research is supported by the National Research Foundation, Singapore, and DSO National Laboratories under the AI Singapore Programme (AISG Award No: AISG2-GC-2023-008), and Career Development Fund (CDF) of Agency for Science, Technology and Research (A*STAR) (No.: C233312028). This work is supported in part by funding from the Canada First Research Excellence Fund as part of the University of Alberta's Future Energy Systems research initiative, Canada CIFAR AI Chairs Program, the Natural Sciences and Engineering Research Council of Canada (NSERC No.RGPIN-2021-02549, No.RGPAS-2021-00034, No.DGECR-2021-00019); as well as JST-Mirai Program Grant No.JPMJMI20B8, JSPS KAKENHI Grant No.JP21H04877, No.JP23H03372, and the support from TIER IV, Inc. and Autoware Foundation.

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
