## A APPENDIX

### A.1 TRANSFERABILITY ACROSS DATASETS

In this section, we extend the evaluation of our defense approach, LRR, to additional datasets to investigate its transferability. Specifically, we test LRR's performance on the challenging LaSOT (Fan et al., 2019), NFS (Kiani Galoogahi et al., 2017) and TrackingNet (Muller et al., 2018) datasets. The LaSOT is a large-scale dataset containing 280 videos. On the other hand, the NFS dataset consists of 100 videos captured using a high-speed camera and is divided into two variants: NFS30 and NFS240, which have frame rates of 30 fps and 240 fps, respectively. Additionally, TrackingNet encompasses a diverse set of 511 video sequences, offering a broad range of real-world scenarios to rigorously evaluate tracking algorithms.

Table 7: Results of LRR defense over four extended datasets.

| SiamRPN++ | Attacks | LaSOT Prec. (%) | | NFS30 Prec. (%) | | NFS240 Prec. (%) | | TrackingNet Prec. (%) | |
|---|---|---|---|---|---|---|---|---|---|
| | | Org. | LRR | Org. | LRR | Org. | LRR | Org. | LRR |
| Res50 | wo.Atk | **48.8** | 48.7 | **59.9** | 56.0 | **71.2** | 69.9 | **69.4** | 67.7 |
| | RTAA | 20.5 | **46.3** | 22.4 | **56.5** | 37.4 | **69.6** | 13.9 | **65.7** |
| | IoUAttack | 39.6 | **46.5** | 42.0 | **55.5** | 65.3 | **67.8** | 61.9 | **65.9** |
| | CSA | 17.5 | **45.3** | 19.6 | **58.0** | 33.5 | **70.5** | 39.7 | **66.5** |
| | SPARK | 19.6 | **48.5** | 40.5 | **59.3** | 16.2 | **70.6** | 29.6 | **55.1** |
| MobileV2 | wo.Atk | **44.8** | 44.0 | **57.2** | 55.8 | **69.0** | 66.7 | 63.6 | **63.9** |
| | RTAA | 12.5 | **40.7** | 16.8 | **55.0** | 25.3 | **68.1** | 5.2 | **62.5** |
| | IoUAttack | 29.6 | **41.4** | 30.7 | **47.8** | 55.8 | **66.3** | 54.1 | **60.8** |
| | CSA | 11.3 | **37.8** | 17.6 | **55.6** | 21.4 | **64.7** | 30.2 | **59.5** |
| | SPARK | 10.2 | **43.9** | 19.5 | **56.8** | 7.0 | **66.5** | 17.1 | **54.1** |

From Table 7, we can observe that our LRR exhibits excellent transferability over large-scale datasets. It successfully defends against adversarial tracking attacks across these challenging datasets.

### A.2 DETAILED ANALYSIS OF LRESAMPLENET'S IMPACT

To validate the effectiveness of our primary contribution, we conducted experiments to demonstrate the influence of the LResampleNet in LRR. Given the independent training of STIR and LResampleNet, STIR should be capable of estimating the perturbation by using spatial-temporal information. We evaluated STIR without resampling and assessed performance on clean data and four attackers across three datasets on two trackers from the SiamRPN++ family. In Table 2, we present the increase in precision for the OTB100 and UAV123 datasets and the rise in EAO value for VOT2019. The results indicate that tracking outcomes without the LResampleNet are less effective than LRR in defending against adversarial tracking attacks. However, it has been observed that using STIR alone causes less damage to the clean data when compared to the LRR approach. This suggests that LRR has the potential to damage clean data. Nevertheless, considering the overall results, an accuracy of less than 2% for OTB100 and UAV123, or an EAO of 0.01 for VOT2019 can be deemed acceptable, considering the enhanced robustness defense capability that LRR offers.

### A.3 VISUALIZATION INSIGHTS

Given the template of the object of interest and an incoming frame, a tracker (e.g., SiamRPN++) aims to predict the object's position by correlating the deep features of both the template and the frame. An attack introduces adversarial perturbation to the frame with the intent to mislead the correlation process. We illustrate the comparison of LRR with and without defense visually in Section 4.2 at the image level, and we aim to delve deeper into the comparison at the correlation level.

More specifically, we provide visualizations in Figure 4 that demonstrate correlation maps from frames processed by our LRR method align much more closely with those unmarred by attack than other defense approaches. As the visualization illustrates, our LRR exhibits lower correlation map differences than DISCO (Ho & Vasconcelos, 2022) and STIR. This is because LRR effectively achieves semantic consistency between the reconstructed frame and the object template, while DISCO and STIR are primarily designed for image quality restoration and overlook the semantic consistency of the template. We observe that DISCO and STIR maintain relatively lower

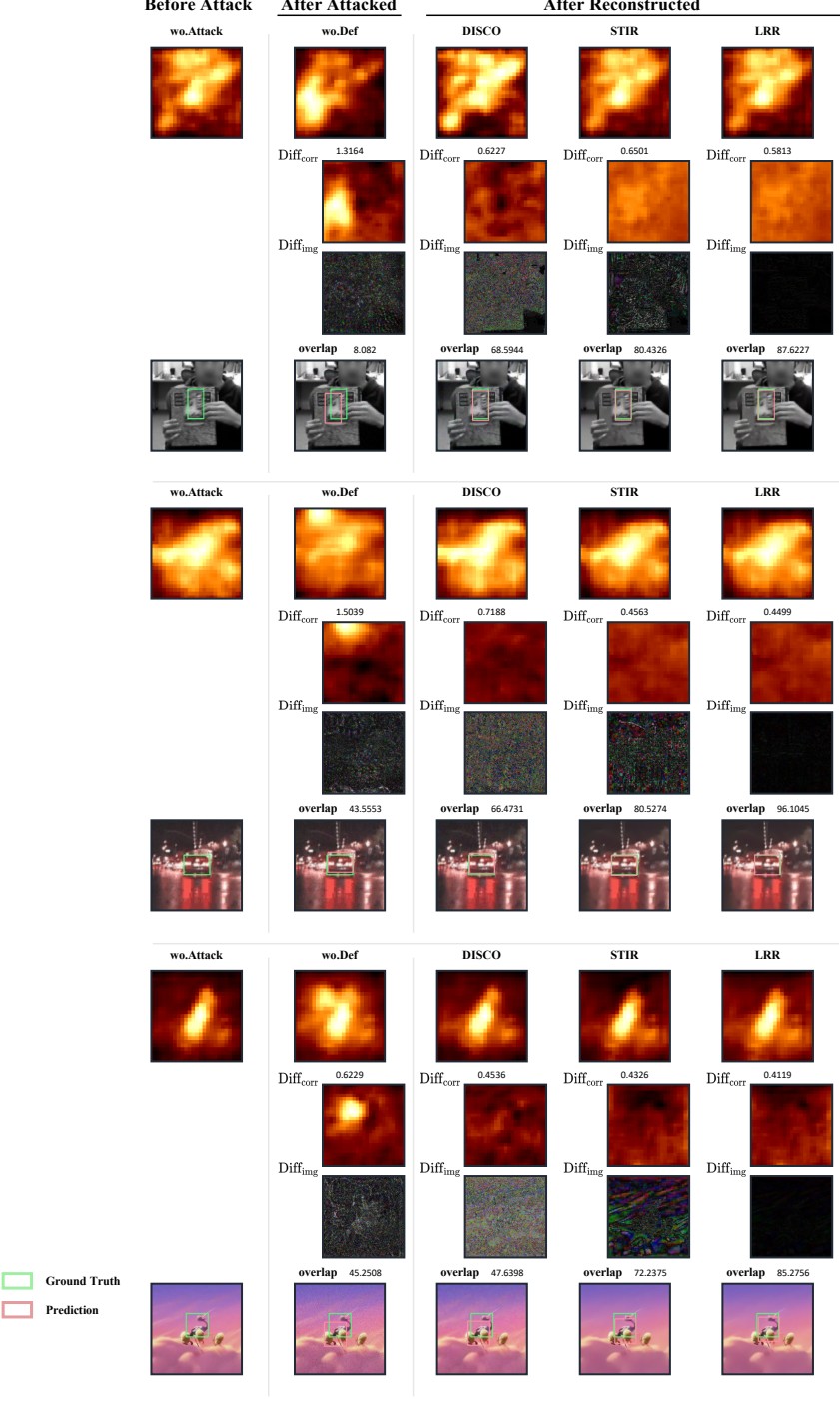

Figure 4: Visualization comparison before & after defense from DISCO, STIR and LRR.

overlap with the ground truth compared to our LRR, highlighting the efficacy and precision of the LRR method in maintaining semantic integrity amidst adversarial perturbations.

Furthering our exploration, we investigate the impact of the resampling module in STIR, informed by the guidance of text embedding through visualization. the LResampling module does not change the features of input frames directly but changes the rendering process of STIR. To elaborate, STIR can reconstruct the colors of any given coordinates as equation 4. Naively, we feed STIR with grid

Table 8: Comparing DISCO, STIR, LRR wo.Lang. and LRR under Four Attacks.

| SiamRPN++ | Defends | OTB100 Prec. (%) | | | | | VOT2019 EAO | | | | | UAV123 Prec. (%) | | | | |
|---|---|---|---|---|---|---|---|---|---|---|---|---|---|---|---|---|
| | | Cln. | RTAA | IoU | CSA | SPARK | Cln. | RTAA | IoU | CSA | SPARK | Cln. | RTAA | IoU | CSA | SPARK |
| Res50 | wo.Def | **91.4** | 32.7 | 75.9 | 47.2 | 69.8 | **0.277** | 0.080 | 0.153 | 0.089 | 0.079 | 79.5 | 41.2 | 70.5 | 46.5 | 40.8 |
| | DISCO | 86.0 | 86.3 | 78.6 | 83.6 | 85.7 | 0.249 | 0.244 | 0.190 | 0.204 | 0.248 | 79.1 | 76.8 | 75.9 | 77.7 | 76.0 |
| | STIR | 88.1 | 85.9 | 84.0 | 85.9 | 87.7 | 0.268 | 0.247 | 0.213 | 0.219 | 0.256 | 79.5 | 77.0 | 78.2 | 79.7 | 77.9 |
| | LRR wo.Lang | 88.1 | 86.1 | 84.4 | 86.0 | 87.8 | 0.270 | 0.250 | **0.219** | 0.222 | 0.254 | **79.6** | 77.4 | 78.4 | 79.8 | 78.1 |
| | LRR | 87.8 | **86.9** | **85.3** | **89.4** | **89.3** | 0.262 | **0.255** | 0.217 | **0.237** | **0.269** | 79.3 | **77.7** | **78.6** | **81.8** | **79.3** |
| MobileV2 | wo.Def | 85.5 | 25.6 | 67.8 | 40.9 | 32.2 | **0.267** | 0.062 | 0.125 | 0.083 | 0.037 | 80.3 | 39.3 | 66.2 | 42.2 | 22.5 |
| | DISCO | 82.9 | 78.7 | 75.0 | 79.9 | 80.1 | 0.175 | 0.161 | 0.132 | 0.166 | 0.208 | 74.6 | 76.2 | 73.3 | 72.9 | 75.3 |
| | STIR | 85.5 | 80.7 | 79.0 | 79.8 | 83.7 | 0.238 | 0.204 | 0.163 | 0.216 | 0.225 | 78.4 | 77.7 | 74.9 | 75.7 | 78.2 |
| | LRR wo.Lang | **86.1** | 81.2 | 79.0 | 81.1 | 84.5 | 0.238 | 0.201 | 0.151 | 0.219 | 0.228 | **79.6** | 78.0 | 75.3 | 76.0 | 78.4 |
| | LRR | 85.6 | **83.2** | **82.1** | **83.9** | **85.4** | 0.240 | **0.205** | **0.166** | **0.223** | **0.239** | 79.1 | **78.7** | **75.9** | **76.2** | **79.0** |

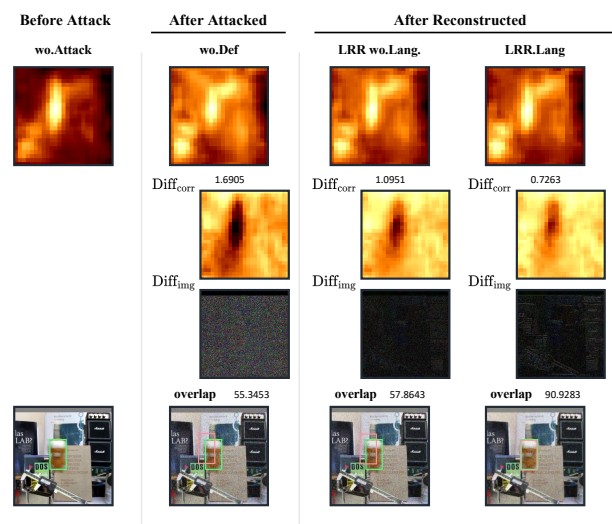

Figure 5: Visualization comparison ResampleNet with & without language guidance when the input frame contains the object of interest.

coordinates (discrete integral coordinates). In this context, we employ the LResampleNet to predict coordinate offsets (non-integral values) around the grid coordinates as illustrated in equation 5, and subsequently, STIR can render an image based on the predicted offsets and grid coordinates. LResampleNet carries dual advantages:

**If the input frame contains the object of interest,** the predicted coordinates are based on the text embedding and can highlight the object automatically. As the visualization of correlation maps shown in Figure 5, our final version could suppress noises from adversarial perturbations. Without language guidance, the overlap of LRR wo.Lang's results with respect to the ground truth is much smaller than the LRR .Lang (57.86 vs. 90.93).

**If the input frame does not contain the object of interest,** the predicted coordinates are around the grid coordinates and will keep the high restoration quality. As shown in Figure 6, LRR could recover the quality and make the prediction similar to the frame without attack.

Moreover, in Table 8, with our LResampleNet, two variants of LRR outperform STIR and DISCO on all datasets and attacks.

Furthermore, we provide the visualizations of clean frames, adversarial frames after attacks, and reconstructed frames after defense, validating the effectiveness of our method in terms of image quality variations. In particular, we consider three typical attacks, *i.e.*, CSA, IoUAttack, and SPARK, respectively, and show their results in Figure 7, Figure 8, and Figure 9. From the visualization results, we observe that: *First*, The three attacks can generate adversarial perturbations with different

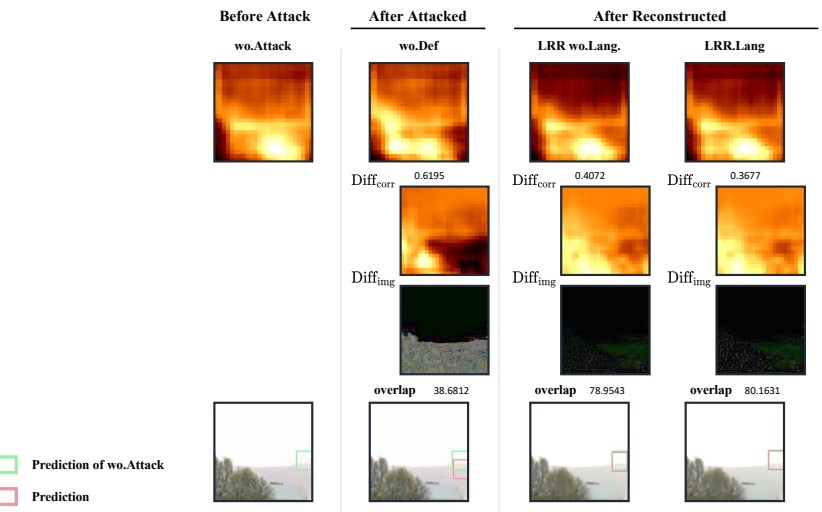

Figure 6: Visualization comparison ResampleNet with & without language guidance when the input frame does not contain the object of interest.

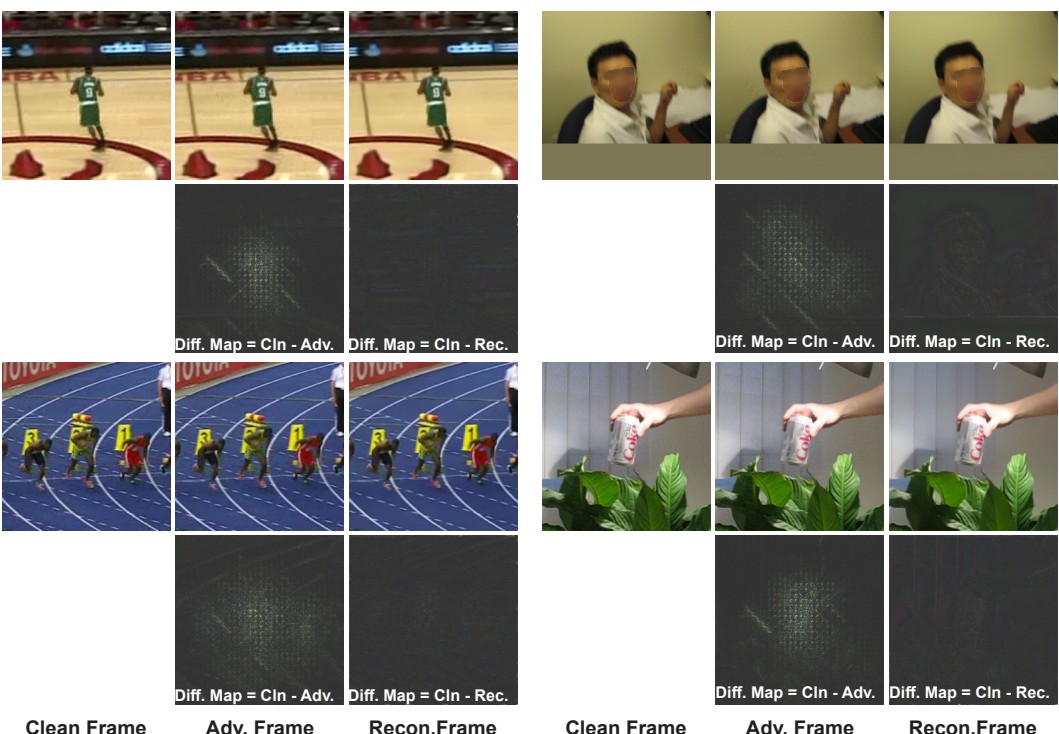

Figure 7: Visualization of clean frames, adversarial frames from CSA, and reconstructed adversarial frames based on our method. We also show the difference map between the adversarial frame and the corresponding clean frame and the difference map between the reconstructed adversarial frame and the corresponding clean frame

textures according to the difference maps shown in the figures. *Second*, for all attacks, our method can eliminate all adversarial perturbations effectively, though they have different textures.

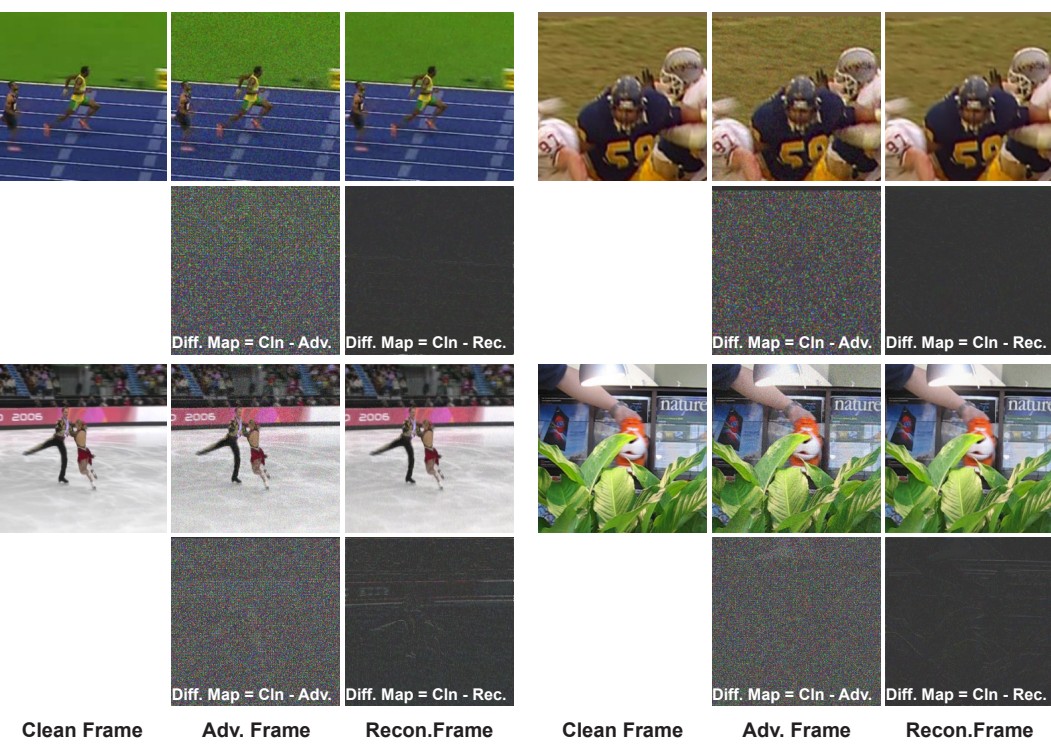

Figure 8: Visualization of clean frames, adversarial frames from IoUAttack, and reconstructed adversarial frames based on our method. We also show the difference map between the adversarial frame and the corresponding clean frame and the difference map between the reconstructed adversarial frame and the corresponding clean frame

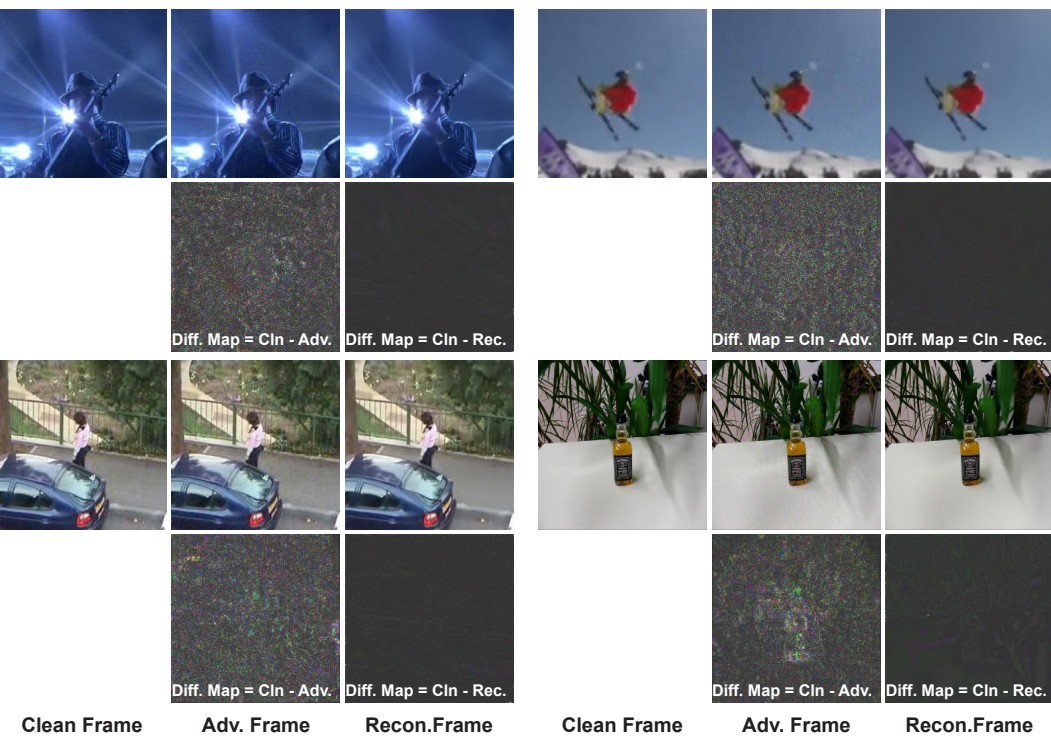

Figure 9: Visualization of clean frames, adversarial frames from SPARK, and reconstructed adversarial frames based on our method. We also show the difference map between the adversarial frame and the corresponding clean frame and the difference map between the reconstructed adversarial frame and the corresponding clean frame

## A.4 TRANSFERABILITY TO TRANSFORMER-BASED TRACKERS

To demonstrate the transferability of our LRR approach, we modified it for the recently proposed ToMP-50 transformer-based tracker model (Mayer et al., 2022), using RTAA to attack and applying LRR for defense, and assessed the outcomes across three different datasets. As substantiated in Table 5, several observations are apparent: firstly, the application of RTAA notably degrades the accuracy of the transformer-based tracker across all three datasets. Secondly, despite these aggressive attacks, our method retains its robust defense capabilities, maintaining high tracking accuracy. This illustrates the notable transferability of the LRR approach, maintaining its effectiveness even when applied to newly developed tracking models, including those based on transformer architectures.

## A.5 COMPARING WITH TRACKING ADVERSARIAL DEFENSE

Table 9: Comparison of LRR with RTAA's Defense on four datasets.

| SiamRPN++ | Attacks | OTB100 Prec. (%) | | | VOT2019 EAO | | | UAV123 Prec. (%) | | | NFS30 Prec. (%) | | |
|---|---|---|---|---|---|---|---|---|---|---|---|---|---|
| | | Org. | LRR | RTAA$_{Def}$ | Org. | LRR | RTAA$_{Def}$ | Org. | LRR | RTAA$_{Def}$ | Org. | LRR | RTAA$_{Def}$ |
| Res50 | wo.Atk | 91.4 | **87.8** | 72.0 | 0.277 | **0.262** | 0.197 | 79.5 | **79.3** | 65.2 | 59.9 | **56.0** | 46.6 |
| | RTAA | 32.7 | **86.9** | 76.5 | 0.080 | **0.255** | 0.155 | 41.2 | **77.7** | 71.8 | 24.4 | **56.5** | 40.5 |
| | IoUAttack | 75.9 | **85.3** | 56.1 | 0.153 | **0.217** | 0.136 | 70.5 | **78.6** | 57.0 | 42.0 | **55.5** | 27.0 |
| | CSA | 47.2 | **89.4** | 31.6 | 0.089 | **0.237** | 0.079 | 46.5 | **81.8** | 41.4 | 19.6 | **58.0** | 13.1 |
| | SPARK | 69.8 | **89.3** | 60.6 | 0.079 | **0.269** | 0.078 | 40.8 | **79.3** | 47.2 | 40.5 | **59.3** | 35.9 |

Table 10: Comparison of LRR with RTAA's Defense cost on OTB100.

| Module | Defense cost per frame (ms) |
|---|---|
| LRR | 39 |
| RTAA$_{Def}$ | 215 |

We have already demonstrated the effectiveness of our proposed method compared to various defense strategies; however, exploration of other defense approaches specifically designed for visual object tracking tasks remains pending. In this section, comprehensive comparisons with RTAA (Jia et al., 2020) are included across four datasets and against four attack methods, as illustrated in Table 9. The RTAA defense method was implemented utilizing the codes from the official repository to defend against the aforementioned attacks.

Clearly, our method presents several notable advantages over RTAA's defense strategy. Firstly, our method consistently achieves superior tracking accuracy compared to RTAA's defense method against all types of attacks and across all datasets examined. Secondly, the impact of our method on clean data is minimal, preserving the integrity and accuracy of the unaffected data. In contrast, RTAA's defense method could notably diminish accuracy when applied to clean data. Additionally, a comparative analysis of the time costs between LRR and RTAA on OTB100 is provided in Table 10. This comparison elucidates the enhanced efficiency of our method over RTAA, strengthening the argument for its application in practical, time-sensitive scenarios. The methodical implementation and rigorous evaluation underscore the robustness and reliability of our method, validating its potential as a superior defense mechanism in visual object-tracking tasks.

## A.6 DETAILED DISCUSSION OF DEFENSE EFFICIENCY

In Section 4.2, we report both the time costs of our methods and the attack costs of the attackers in Table 6, respectively. We demonstrate that our proposed methods exhibit superior frame processing efficiency compared to most attackers, with the exception of CSA (Yan et al., 2020), which employs a fast perturbation generator. Furthermore, our LRR surpasses STIR in adversarial attack defense capability, sacrificing only a negligible amount of efficiency—4ms per input frame defense. In the case of less efficient attackers such as IoU Attack (Jia et al., 2021) and RTAA (Jia et al., 2020), we receive attacked frame sequences at a rate of less than 0.1 frames per second (fps). In this context, the computational cost added by LRR is practically negligible. For more efficient attackers, such as SPARK (Guo et al., 2020b) and CSA, under the assumption that the attacker and defender utilize the same computational resources, our LRR method trades off a portion of tracking efficiency in favor

of a significant improvement in the tracker's robustness. In real-world scenarios, where attackers and defenders are typically deployed on separate systems, our STIR defense sustains online frame processing at an approximate rate of 29 fps, while LRR functions at around 25 fps.

Moreover, computation time costs can be further optimized by adapting the defense policy. For instance, by employing the target overlap ratio compared to the previous frame as a threshold, we can bypass processing for 25% of frames and still maintain an overlap ratio not lower than 85%.

## A.7 FEASIBILITY OF USING DIFFUSION FOR TRACKING DEFENSE

We explore the efficacy of the recently developed diffusion-based adversarial defense method, Diff-Pure (Nie et al., 2022), for tracking defense. Specifically, we apply DiffPure to safeguard against three attacks, i.e., RTAA, IoUAttack, and CSA, that are used to attack the SiamRPN++ Res50 tracker on the OTB100 dataset. In our empirical study, we use DiffPure's default parameters for defense but vary the number of iterative time steps (i.e., T=1, 10, 50).

Table 11 illustrates that the three DiffPure variants enhance the precision of the tracker under different attacks, albeit to a lesser extent compared to our approach, LRR. Notably, DiffPure(T=50) is 86.9 times slower than LRR, requiring an average of 3391 ms for each frame, rendering it nearly impractical for tracking tasks. Even with a reduced time step to 1, DiffPure speeds up to 146 ms per frame, still 3.7 times slower than LRR. It is essential to note that the default DiffPure configuration sets T=100 time steps for purification, which is impractical for tracking tasks due to time constraints. In conclusion, further investigation is needed to understand the potential of leveraging diffusion for tracking defense.

Table 11: Comparing DiffPure Nie et al. (2022) with LRR on OTB100 where we use them to defend CSA and RTAA for the SiamRPN++ Res50.

| Defense method | Cln. | RTAA | IoUAttack | CSA | Time (ms) |
|---|---|---|---|---|---|
| w.o. Defense | 91.4 | 32.7 | 75.9 | 47.2 | - |
| LRR | 87.8 | **86.9** | **85.3** | **89.4** | 39 |
| DiffPure(T=1) | 87.9 | 52.3 | 78.5 | 83.5 | 146 |
| DiffPure(T=10) | 88.1 | 53.7 | 78.8 | 84.1 | 742 |
| DiffPure(T=50) | 88.2 | 54.2 | 79.0 | 84.3 | 3391 |

## A.8 COMPARING WITH RESIZING AND COMPRESSION-BASED DEFENSES

We implemented a resizing-based defense using the 'cv.resize' operation in OpenCV. Specifically, for an input image $\mathbf{I} \in R^{H \times W}$, we first downsample it by a factor $r$ and get image $\mathbf{I}_\downarrow \in R^{rH \times rW}$. Then, we upsample it to the raw resolution, generating the reconstructed image $\hat{\mathbf{I}} \in R^{H \times W}$. Following this, we input the reconstructed images into trackers.

To assess the effectiveness of resizing-based defense, we varied the downsampling ratio within the range $r \in \{0.9, 0.8, \ldots, 0.1\}$. As shown in Table 12, we observe that: 1. Resizing proves to enhance the tracker's accuracy under various attacks. 2. The efficacy of this enhancement varies depending on the attack type. Resizing significantly mitigates the impact of the SPARK attack, elevating precision from 69.8 to 83.9, but exhibits limited effectiveness against the RTAA, where precision increases modestly from 32.7 to 49. 3. The influence on RTAA remains constrained as precision increases from 32.7 to 49.3. Gradually increasing $r$ improves precision under RTAA but adversely affects precision in clean data and IoUAttack scenarios. 4. Compared to the resizing method, LRR consistently improves tracker precision across all attacks, showcasing a noteworthy advantage while maintaining a high score on clean data.

Regarding compression, we utilize JPG compression for image reconstruction, adjusting compression qualities with $q \in [98\%, 96\%, 94\%, 92\%, 90\%]$. The results are presented in Table 13, and the following observations are made: 1. Compression with a high-quality requirement exhibits limited influence on various attacks. 2. As the compression quality decreases, precision on different attacks increases, highlighting the effectiveness of compression as a defense mechanism against adversarial

Table 12: Comparison of resizing-based defense with different settings of $r$ on OTB100.

| SiamRPN++ | Attacks | OTB100 Prec. (%) | | | | | | | | | |
|---|---|---|---|---|---|---|---|---|---|---|---|
| | | Org. | LRR | $r = 0.9$ | $r = 0.8$ | $r = 0.7$ | $r = 0.6$ | $r = 0.5$ | $r = 0.4$ | $r = 0.3$ | $r = 0.2$ | $r = 0.1$ |
| Res50 | wo.Atk | 91.4 | 87.8 | 86.5 | 86.2 | 85.9 | 85.4 | 85.3 | 82.7 | 82.7 | 80.8 | 69.9 |
| | RTAA | 32.7 | 86.9 | 49.3 | 56.7 | 62.0 | 69.0 | 72.5 | 77.3 | 80.2 | 80.3 | 69.0 |
| | IoUAttack | 75.9 | 85.3 | 80.3 | 80.1 | 79.0 | 79.0 | 76.1 | 76.5 | 74.9 | 72.0 | 63.3 |
| | CSA | 47.2 | 89.4 | 71.6 | 80.3 | 84.6 | 86.0 | 83.8 | 83.1 | 83.0 | 81.8 | 69.1 |
| | SPARK | 69.8 | 89.3 | 83.9 | 85.1 | 88.1 | 87.8 | 86.0 | 87.7 | 85.4 | 82.5 | 72.2 |

Table 13: Comparison of compression-based defense with different settings of $q$ on OTB100.

| SiamRPN++ | Attacks | OTB100 Prec. (%) | | | | | | |
|---|---|---|---|---|---|---|---|---|
| | | Org. | LRR | $q = 98\%$ | $q = 96\%$ | $q = 94\%$ | $q = 92\%$ | $q = 90\%$ |
| Res50 | wo.Atk | 91.4 | 87.8 | 90.8 | 89.6 | 90.2 | 89.7 | 90.1 |
| | RTAA | 32.7 | 86.9 | 33.5 | 42.9 | 50.1 | 60.6 | 66.1 |
| | IoUAttack | 75.9 | 85.3 | 74.8 | 74.4 | 76.5 | 76.2 | 77.1 |
| | CSA | 47.2 | 89.4 | 49.0 | 51.4 | 51.8 | 56.3 | 58.7 |
| | SPARK | 69.8 | 89.3 | 78.1 | 82.1 | 83.6 | 85.4 | 85.9 |

tracking. 3. The improvements achieved by compression under attacks are limited and fall short of the results obtained with LRR.

## A.9 DETAILS OF ADVERSARIAL TRACKING ATTACKS

We implement adversarial tracking attacks via the released codes from existing tracking adversarial attacks (*i.e.*, RTAA (Jia et al., 2020), IoUAttack (Jia et al., 2021), CSA Yan et al. (2020), and SPARK Guo et al. (2020b)) to implement attacks in our experiments. We detail some setups as follows.

For RTAA, we utilized their originally released code (`https://github.com/VISION-SJTU/RTAA/blob/main/DaSiamRPN/code/run_attack.py`). The process follows these steps: 1. RTAA receives an incoming image and the target location where the image is the search region cropped by the studied tracker. 2. RTAA adds adversarial perturbations to the search region and outputs an adversarial example for the tracker to handle. At each frame, the attack optimizes the adversarial perturbation iteratively ten times, with the maximum perturbation set to 10/255. 3. RTAA outputs the optimized adversarial example as the new search region.

For the IoU Attack, we adhered to their default setups in their released code for conducting our experiments (`https://github.com/VISION-SJTU/IoUattack/blob/main/pysot/tools/test_IoU_attack.py`). Specifically, we follow the subsequent steps: 1. IoUAttack receives the frame and targeted bounding box as inputs. 2. IoUAttack optimizes the perturbations iteratively until the IoU score is below the predefined score (See the released code for details). 3. IoU outputs the optimized adversarial frame to attack the tracker.

For CSA, we employed their released pre-trained perturbation generator to attack each frame (`https://github.com/MasterBin-IIAU/CSA/blob/efd69a5705dd21c6701fd4ae7922f3a44647069a/pysot/pysot/tracker/siamrpn_tracker.py`). Specifically, CSA receives the clean search region and feeds it to the pre-trained perturbation generator. Then, the generator outputs the adversarial perturbation added to the clean search region.

In the case of SPARK (`https://github.com/tsingqguo/AttackTracker/blob/main/tools/attack_oim.py`), we employed the targeted attack approach provided in SPARK's default setup from their released code for attacks. The procedure involves the following steps: 1. SPARK takes the search region, cropped from the input frame, the targeted trajectory, and the targeted tracker as inputs. 2. SPARK optimizes the perturbations, iterating 10 times every 30 frames and 2 times at other frames. The maximum perturbation allowed is 0.3. 3. SPARK generates the optimized adversarial search region to attack the tracker.