# OpenReview forum: "LRR: Language-Driven Resamplable Continuous Representation against Adversarial Tracking Attacks"
_ICLR.cc/2024/Conference — ICLR 2024 poster_

### Official Review · Reviewer_DMWE · 2023-10-31

**Soundness:** 2 fair
**Presentation:** 2 fair
**Contribution:** 2 fair
**Rating:** 5
**Confidence:** 5

**Summary:**

This work presents a study on improving object tracking performance on adversarial data while maintaining the model's superiority over clean data. The essence is building a spatial-temporal implicit representation using the semantic text guidance of the object of interest extracted from the language-image model. Then the novel representation is used to reconstruct incoming frames. Experimental results on different benchmarks show the effectiveness of the proposed framework. Overall, the paper sounds reasonable.

**Strengths:**

- The writing is clear and easy to follow.
- Introducing language embedding to adversarial training is interesting.
- Reconstructing video frames to defend against adversarial attacks is reasonable.
- Experimental results are sufficient.

**Weaknesses:**

- Although it's interesting to introduce nlp embedding in the adversarial defense framework, it sounds not so reasonable. Using a template seems enough for the purpose. Besides, single-object tracking is class-agnostic. What if the object class is unknown? How to run the proposed model in this situation?
- Lack of experiments on more challenging benchmarks, e.g. trackingnet, tnl2k (Towards More Flexible and Accurate Object Tracking with Natural Language: Algorithms and Benchmark).
- It is noticed that only SiamRPN++ is used as the baseline model. The baseline model is out of date. Do the effectiveness and conclusion still hold on recent transformer-based models, like OSTrack, SwinTrack, MixFormer? What if the tracking model is already able to track objects using natural language like the algorithms introduced in tnl2k.

**Questions:**

Please conduct experiments on recent trackers and provide results on more challenging and up-to-date benchmarks.

---

> ### Author Response · Authors · 2023-11-22
> **Q1: The reasonability of our method: our method does not require knowing the category of the object.**
>
> Thank you for the comments. It appears that there might be some misunderstanding regarding certain details in the submission. The language utilized in Sec 3.3 is not predetermined; rather, it is extracted from the provided template using the CLIP model. Specifically, we initiate the process by employing CLIP's image encoder to extract the template's embedding. Subsequently, with a set of texts encompassing potential categories of the object, we compare the template's embedding with the embeddings of all the texts. Following this, we select the text embedding that exhibits the highest similarity with the template's embedding as the $\mathbf{z}_\text{txt}$ used in Eq. (6). Note that the text set can be updated based on different application scenarios. Alternative vision-language models or image caption methods can also achieve the same objective.
>
> To address the concern, we have added an experiment by using the BLIP to extract the text embedding $\mathbf{z}_\text{txt}$. As results are shown in the following table, we observe that the two methods achieve similar results.
>
> Table R-1: Using CLIP and BLIP to extract the text embeddings.
>
> | OTB100 (Precision)  | w.o. Defense | **LRR with CLIP** | **LRR with BLIP** |
> | -------------------    | --------------- | -------| -------|
> |Clean                   | 91.4	|  87.8	   |  88.4 |
> |RTAA                    | 32.7	|  86.9	   |  86.4 |
> |IoU                     | 75.9	|  85.3	   |  84.9 |
> |CSA                     | 47.2	|  89.4	   |  87.5 |
> |SPARK                   | 69.8 |  89.3	   |  89.0 |
>
>
> To avoid the misunderstanding, we have revised section 3.3 by adding the above discussions in the manuscript to highlight the details.

---

> ### Author Response · Authors · 2023-11-22
> **Q2: Evalution on the TrackingNet and TNL2K datasets.**
>
> Thank you for the suggestions. In the original supplementary material, we have reported the results on other three larger datasets, i.e., LaSOT, NFS30, and NFS240, in Appendix A.1 with Table 7.
>
> Following the provided suggestions, we have conducted additional experiments on the TrackingNet and TNL2K datasets to validate the effectiveness of our approach.
>
> For the TrackingNet dataset, we initiated four attacks (RTAA, CSA, IoUAttack, and SPARK) against the SiamRPN++ Res50. Subsequently, we employed our LRR to defend against these attacks. As illustrated in Table R-2, the four attacks substantially diminished the tracker's accuracy, and LRR demonstrated its capability to enhance the tracker's robustness under diverse attack scenarios.
>
>
> Table R-2: SiamRPN++ ResNet50 Precision (%) on TrackingNet under four attacks.
>
> | TrackingNet (Precision) | w.o. Defense | LRR  |
> | ---------------------------------- | ------------ | ---- |
> | wo.Atk                             | 69.4         | 67.7 |
> | RTAA                               | 13.9         | 65.7 |
> | IoUAttack                          | 61.9         | 65.9 |
> | CSA                                | 39.7         | 66.5 |
> | SPARK                              | 29.6         | 55.1 |
>
> We have added the results to Table 7 in Appendix A.1 and added explanations in Sec 4.2.
>
>
> In relation to the TNL2K dataset, it comprises 700 sequences for testing, with 100 sequences specifically featuring adversarial examples generated using the RTAA. However, upon executing the SiamRPN++ on the provided 100 adversarial sequences, we observed a limited impact on the tracking accuracy. To mitigate any potential risks of misinterpretation, we conducted attacks on the remaining 600 sequences using the three attacks (i.e., RTAA, CSA, and SPARK) for a fair comparison, followed by employing our LRR for defense. Please note that IoUAttack is excessively time-consuming and cannot be executed within the given constraints. We plan to include it once the initial set of experiments is completed. As indicated in the results presented in Table R-3, our observations align closely with the findings from the TrackingNet experiments.
>
>
> Table R-3: SiamRPN++ ResNet50 Precision (%) on TrackingNet under three attacks.
>
> |        | w.o. Dfense | LRR  |
> | ------ | ----------- | ---- |
> | wo.Atk | 57.3        | 54.2 |
> | RTAA   | 16.3        | 50.9 |
> | CSA    | 32.9        | 56.0 |
> | SPARK  | 33.7        | 56.9 |

---

> ### Author Response · Authors · 2023-11-22
> **Q3: Experiments on transformer-based tracker.**
>
> Thank you for the comments. The reviewer might neglect the results in Table 5 and the corresponding discussions. Here, we would like to clarify further the reasons for using SiamRPN++ as the main experimental setup and report the results on a state-of-the-art transformer method, i.e., ToMP-50 [Mayer et al. 2022].
>
> Specifically, the main objective of this submission is to explore an effective way against SOTA adversarial tracking attacks. We select SiamRPN++ trackers [Li et al.,2019] as the main study subjects due to the following reason: all of the representative attacks including IoUAttack, SPARK, CSA, and RTAA have official codes or models to attack SiamRPN++ trackers. By conducting experiments on SiamRPN++ trackers, we can avoid the performance reduction issues caused by the re-implementation of different attacks and compare defense methods effectively.
>
> To address the concerns, we highlight experiments on a transformer-based tracker in Table 5, i.e., ToMP-50 [Mayer et al. 2022]. Specifically, we use the RTAA to attack the ToMP-50 and use our method LRR for defense, and evaluate the results on three datasets. As shown in the following table, we see that: **Firstly**, RTAA can also lead to obvious accuracy reductions for the transformer-based tracker on three different datasets. **Secondly**, our method still presents powerful defense capability and keep high or even higher tracking accuracy than the results on clean data.
>
> To address the concerns more comprehensively, our intention was to include the mentioned trackers in our evaluation. However, due to time constraints, we were only able to obtain results for OSTracker using its official code, as presented in Table R-4. It is worth noting that adapting existing attacks to new trackers would require a significant amount of time.
>
> Given that ToMP-50 and the mentioned trackers are papers from the same time period, we believe the conclusions drawn demonstrate the effectiveness of our method on transformer-based trackers. We will involve the results of these trackers in our future revision. Here, we have added a discussion about this in Sec 2.
>
>
> Table 5: Defense results on ToMP-50 across three datasets.
>
> | Results on OTB100 (Precision)-  | Without Defense | **LRR(Ours)** |
> | -------------------    | --------------- | -------|
> |Clean                   | 90.1	| **90.5** |
> |RTAA                    | 61.8 | **91.2** |
>
> | Results on VOT2019 (EAO)------| Without Defense | **LRR(Ours)** |
> | -------------------    | --------------- | -------|
> |Clean                   | 0.549	| **0.535** |
> |RTAA                    | 0.352 | **0.542** |
>
> | Results on UAV123 (Precision)-- | Without Defense | **LRR(Ours)** |
> | -------------------    | --------------- | -------|
> |Clean                   | 88.2	| **88.8** |
> |RTAA                    | 58.5 | **89.2** |
>
>
> Table R-4: Defense results on OSTracker across one dataset.
>
> | Results on OTB100 (Precision)-  | Without Defense | **LRR(Ours)** |
> | -------------------    | --------------- | -------|
> |Clean                   | 91.7	| **90.4** |
> |IoUAttack                    | 75.5 | **89.6** |
>
>
> [Mayer et al. 2022] C. Mayer, M. Danelljan, G. Bhat, M. Paul, D. P. Paudel, F. Yu, L. Van Gool. Transforming model prediction for tracking. in CVPR 2022.

---

> ### Author Response · Authors · 2023-11-22
> **Q4: Why not involve the natural language-based trackers.**
>
> Thank you for the suggestions. It's important to note that natural language-based trackers have distinct designs and pipelines compared to template-based tracking methods. Despite this distinction, all existing attacks are tailored for traditional template-based trackers. We have observed that transferring these existing attacks to natural language-based trackers is not straightforward. We intend to delve into this challenge as part of our future work and to address this, we have incorporated a discussion in the conclusion section.
>
> [Wang et al. 2021] X. Wang, X. Shu, Z. Zhang, B. Jiang, Y. Wang, Y. Tian, & F. Wu (2021). Towards more flexible and accurate object tracking with natural language: Algorithms and benchmark. in CVPR, 2021.

---

### Official Review · Reviewer_n1Fj · 2023-11-02

**Soundness:** 3 good
**Presentation:** 2 fair
**Contribution:** 3 good
**Rating:** 6
**Confidence:** 4

**Summary:**

The authors propose a method to defend visual object tracking against adversarial attacks. They introduce a spatial-temporal implicit representation (STIR) that constructs neighboring pixels, and a language-driven resample network (LResampleNet) that provides consistency between reconstructed frames and object templates. They use the CLIP model to guide their approach. The effectiveness of their method is demonstrated through experiments on the OTB100, VOT2019, and UAV123 datasets, which show that it can effectively defend against recent VOT attack methods.

**Strengths:**

1. The proposed method considers both the spatial and temporal information during defense, which is reasonable.
2. Using the language-image model to guide the defense process is interesting.
3. The experiments are thorough, including the defense results against various attack methods, as well as ablation studies on the efficiency of each component.

**Weaknesses:**

1. Provide visual comparisons among clean images, adversarial images, and the image after defense to show the visual effects and their difference. The tracking results should be added as well.
2. For the experiments, the reviewer suggests that the authors compare with basic defense methods, including adversarial training and image preprocessing (e.g., resize or compression). And analyze the pros and cons between the proposed method and the defense methods mentioned above.
3. In Table 1, for the results without defense, how to implement the attack is not clear. In other words, which attack method is selected for the results in Table 1?
4. There are some minor problems, and the author should polish this paper again.
- In Table1, defends -> defense?
- In Table 2 and 3, IouAttack -> IoUAttack. It is a typo.

**Questions:**

1. Please supply the visual comparisons among clean images, adversarial images, and the image after defense.
2. Please compare with other basic defense methods, like resizing or compression.
3. Please state the details of the experiments in Table 1.
4. Fix the typos and polish this paper again.

---

> ### Author Response · Authors · 2023-11-22
> **Q1: Please supply the visual comparisons among clean images, adversarial images, and the image after defense.**
>
> Thanks for the comments. In our original submission, we have provided the visualization results of correlation maps and images before attack (i.e., clean examples), after attack (adversarial examples), and after reconstruction (defense examples) in the supplementary material (i.e., Appendix A.2) to validate our performance. To further address the concerns, we add more visualization results with higher resolutions in the supplementary material (i.e., Appendix A.2) with Figure 7, Figure 8, and Figure 9. Moreover, we have added corresponding explanations in the main submission (Sec. 4.1).

---

> ### Author Response · Authors · 2023-11-22
> **Q2: Please compare with other basic defense methods, like resizing, compression, and adversarial training.**
>
> Thanks for the suggestion. We implemented a resizing-based defense using the cv.resize operation in OpenCV. Specifically, for an input image $\mathbf{I}\in R^{H\times W}$, we first downsample it by a factor $r$ and get image $\mathbf{I}_\downarrow\in R^{rH\times rW}$. Then, we upsample it to the raw resolution, generating the reconstructed image $\hat{\mathbf{I}} \in R^{H\times W}$. Following this, we input the reconstructed images into trackers.
>
> To assess the effectiveness of resizing-based defense, we varied the downsampling ratio within the range $r\in \{0.9, 0.8,\ldots,0.1\}$. As shown in Table 12, we observe that:
>
> 1. Resizing proves to enhance the tracker's accuracy under various attacks.
> 2. The efficacy of this enhancement varies depending on the attack type. Resizing significantly mitigates the impact of the SPARK attack, elevating precision from 69.8 to 83.9, but exhibits limited effectiveness against the RTAA, where precision increases modestly from 32.7 to 49.
> 3. The influence on RTAA remains constrained as precision increases from 32.7 to 49.3. Gradually increasing $r$ improves precision under RTAA but has adverse effects on precision in clean data and IoUAttack scenarios.
> 4. In comparison to the resizing method, LRR consistently improves tracker precision across all attacks, showcasing a noteworthy advantage while maintaining a high score on clean data.
>
> Regarding compression, we utilize JPG compression for image reconstruction, adjusting compression qualities with
> $q\in {0.98, 0.96, 0.94, 0.92, 0.90 }$. The results are presented in Table 13, and the following observations are made:
>
> 1. Compression with a high-quality requirement shows limited impact on various attacks.
> 2. With a decrease in compression quality, precision increases across different attacks, underscoring the efficacy of compression as a defense mechanism against adversarial tracking.
> 3. The improvements achieved by compression under attacks are limited and fall short of the results obtained with LRR.
>
> To address the concerns, we have added the above discussion in the supplementary material (Appendix A.8) with explanations in section 4.2.
>
> Note that, in the original submission, we have involved the adversarial training (AT)-based methods as defense baselines in Section 4 and Table 1, which include three variants, i.e., $AT_{FGSM}$, $AT_{PGD}$, $AT_{CSA}$. We have introduced the three methods in the subsection "Defence baselines" on page 6-7.
>
> Table 12: Comparing DiffPure (Nie et al. 2022) with LRR when defending CSA and RTAA.
>
> | OTB100 (Precision) | w.o. Defense | LRR  | r=0.9    | r=0.8 | r=0.7    | r=0.6    | r=0.5 | r=0.4 | r=0.3 | r=0.2    | r=0.1 |
> | ----------------------------- | ------------ | ---- | -------- | ----- | -------- | -------- | ----- | ----- | ----- | -------- | ----- |
> | wo.Atk                        | 91.4         | 87.8 | 86.5 | 86.2  | 85.9     | 85.4     | 85.3  | 82.7  | 82.7  | 80.8     | 69.9  |
> | RTAA                          | 32.7         | 86.9 | 49.3     | 56.7  | 62.0     | 69.0     | 72.5  | 77.3  | 80.2  | 80.3 | 69.0  |
> | IoUAttack                     | 75.9         | 85.3 | 80.3 | 80.1  | 79.0     | 79.0     | 76.1  | 76.5  | 74.9  | 72.0     | 63.3  |
> | CSA                           | 47.2         | 89.4 | 71.6     | 80.3  | 84.6     | 86.0 | 83.8  | 83.1  | 83.0  | 81.8     | 69.1  |
> | SPARK                         | 69.8         | 89.3 | 83.9     | 85.1  | 88.1 | 87.8     | 86.0  | 87.7  | 85.4  | 82.5     | 72.2  |
>
>
>
> Table 13: Comparing DiffPure (Nie et al. 2022) with LRR when defending CSA and RTAA.
>
> |           | w.o. Defense | LRR  | q=98%  | q=96% | q=94% | q=92%| q=90%|
> | ----      | ------------ | ---- | ------ | ----- | ----- | ---- | ---- |
> | wo.Atk    | 91.4         | 87.8 | 90.8   | 89.6  | 90.2  | 89.7 | 90.1 |
> | RTAA      | 32.7         | 86.9 | 33.5   | 42.9  | 50.1  | 60.6 | 66.1 |
> | IoUAttack | 75.9         | 85.3 | 74.8   | 74.4  | 76.5  | 76.2 | 77.1 |
> | CSA       | 47.2         | 89.4 | 49.0   | 51.4  | 51.8  | 56.3 | 58.7 |
> | SPARK     | 69.8         | 89.3 | 78.1   | 82.1  | 83.6  | 85.4 | 85.9 |

---

> ### Author Response · Authors · 2023-11-22
> **Q3: In Table 1, how to implement the attack is not clear**
>
> Thanks for the comments. We follow the released codes from existing tracking adversarial attacks (i.e., RTAA(Jia et al., 2020), IoU(Jia et al., 2021), CSA(Yan et al., 2020), and SPARK (Guo et al., 2020)) to implement attacks in our experiments. We detail some setups as follows. We also add a section in the supplementary material (i.e., Appendix A.9) with the explanations in Sec 4.
>
> For RTAA, we utilized their originally released code (https://github.com/VISION-SJTU/RTAA/blob/main/DaSiamRPN/code/run_attack.py). The process follows these steps: 1. RTAA receives an incoming image and the target location where the image is the search region cropped by the studied tracker. 2. RTAA adds adversarial perturbations to the search region and outputs an adversarial example for the tracker to handle. At each frame, the attack optimizes the adversarial perturbation iteratively ten times, with the maximum perturbation set to 10/255. 3. RTAA outputs the optimized adversarial example as the new search region.
>
> For the IoU Attack, we adhered to their default setups in their released code for conducting our experiments (https://github.com/VISION-SJTU/IoUattack/blob/main/pysot/tools/test_IoU_attack.py). Specifically, we follow the subsequent steps: 1. IoUAttack receives the frame and bounding box as inputs. 2. IoUAttack optimizes the perturbations iteratively until the IoU score is below the predefined score (See the released code for details). 3. IoU outputs the optimized adversarial frame to attack the tracker.
>
> For CSA, we employed their released pre-trained perturbation generator to attack each frame (https://github.com/MasterBin-IIAU/CSA/blob/efd69a5705dd21c6701fd4ae7922f3a44647069a/pysot/pysot/tracker/siamrpn_tracker.py#L220). Specifically, CSA receives the clean search region and feeds it to the pre-trained perturbation generator. Then, the generator outputs the adversarial perturbation that is added to the clean search region.
>
> In the case of SPARK (https://github.com/tsingqguo/AttackTracker/blob/main/tools/attack_oim.py), we employed the targeted attack approach provided in SPARK's default setup from their released code for attacks. The procedure involves the following steps: 1. SPARK takes the search region, cropped from the input frame, the targeted trajectory, and the targeted tracker as inputs. 2. SPARK conducts iterative optimization on the perturbations, iterating 10 times every 30 frames and 2 times at other frames. The maximum perturbation allowed is 0.3. 3. SPARK generates the optimized adversarial search region to attack the tracker.

---

> ### Author Response · Authors · 2023-11-22
> **Q4: Fix the typos and polish this paper again.**
>
> Thanks for the suggestions. We've diligently proofread the entire submission, rectifying some typos and grammar errors.

---

### Official Review · Reviewer_bxnS · 2023-11-02

**Soundness:** 3 good
**Presentation:** 2 fair
**Contribution:** 3 good
**Rating:** 6
**Confidence:** 4

**Summary:**

This paper presents an adversarial defense method against recent attacks on visual object tracking. The defense method is guided by the language-image mode CLIP to reconstruct the area that has been perturbed by attacks. Two modules named the spatial-temporal implicit representation (STIR) and the language-driven resample network (LResampleNet) are involved in the whole framework to obtain a consistent representation. The experimental results against four attack methods are evaluated on three datasets.

**Strengths:**

1. The idea of using an image-language model (CLIP) to defend against adversarial attacks is interesting.
2. The experiments show the effectiveness of the proposed defense against various types of adversarial attacks and can be applied to different trackers (e.g., CNN-based and transformer-based trackers).

**Weaknesses:**

1. The proposed method basically relies on the reconstruction technology to destroy the distribution of perturbations. For VOT, does the proposed method reconstruct the whole search image? An intuitive and simple idea is to apply the inversion technology in StyleGAN or Diffusion on it, does it work? Please give some analysis on this point.
2. Please provides some visual result on the difference between clean images and adversarial examples, and between clean images and defense examples, to show how the distribution of adversarial perturbations is reduced or suppressed.
3. The writing of the paper needs to improve. Some results in supplementary materials can be combined in the main paper to better support the effectiveness of the proposed method.

**Questions:**

1. Please provide some analysis on directly implementing the inversion technology in StyleGAN or Diffusion.
2. Please provide some visual results before and after attacks to intuitively illustrate how the adversarial perturbations are eliminated.
3. Add the key results and analysis in supplementary materials to the main paper.

---

> ### Author Response · Authors · 2023-11-22
> **Q1: Implementing Diffusion for tracking defense**
>
> Thank you for your valuable suggestions. In response, we have incorporated an experiment in Section 4.2, utilizing the recently proposed diffusion-based reconstruction method known as DiffPure (Nie et al. 2022) to assess its efficacy against adversarial tracking attacks. We contend that DiffPure may not be optimal for tracking defense due to its iterative denoising nature and time-consuming characteristics, making it less adaptable to real-time trackers. In our initial submission, we addressed the primary concerns associated with DiffPure in the related work section. To provide further clarification, we have introduced a new subsection in the supplementary material (Appendix A.7) to elaborate on this aspect. Additionally, a discussion has been included in Sec. 2 for a more comprehensive coverage.
>
>
> **Feasibility of using diffusion for tracking defense.**
> We explore the efficacy of the recently developed diffusion-based adversarial defense method, DiffPure (Nie et al., 2022), for tracking defense. Specifically, we apply DiffPure to safeguard against three attacks—RTAA, IoUAttack, and CSA—based on the SiamRPN++ Res50 tracker. In our empirical study, we use DiffPure's default parameters for defense but vary the number of iterative time steps (i.e., T=1, 10, 50).
>
> Table 11 illustrates that the three DiffPure variants enhance the precision of the tracker under different attacks, albeit to a lesser extent compared to our approach, LRR. Notably, DiffPure(T=50) is 86.9 times slower than LRR, requiring an average of 3391 ms for each frame, rendering it nearly impractical for tracking tasks. Even with a reduced time step to 1, DiffPure speeds up to 146 ms per frame, still 3.7 times slower than LRR. It is essential to note that the default DiffPure configuration sets T=100 time steps for purification, which is impractical for tracking tasks due to time constraints. In conclusion, further investigation is needed to understand the potential of leveraging diffusion for tracking defense.
>
>
> Table 11: Comparing DiffPure (Nie et al. 2022) with LRR when defending RTAA, IoUAttack, and CSA.
>
> | Defense Method  | Cln. | RTAA | IoUAttack | CSA  | Time cost per frame (ms) |
> | --------------- | ---- | ---- | --------- | ---- | -------------- |
> | wo. Defense     | 91.4 | 32.7 | 75.9      | 47.2 | -              |
> | LRR (Ours)      | 87.8 | 86.9 | 85.3      | 89.4 | 39             |
> | DiffPure (T=1)  | 87.9 | 52.3 | 78.5      | 83.5 | 146            |
> | DiffPure (T=10) | 88.1 | 53.7 | 78.8      | 84.1 | 742            |
> | DiffPure (T=50) | 88.2 | 54.2 | 79.0      | 84.3 | 3391           |

---

> ### Author Response · Authors · 2023-11-22
> **Q2: More visualization results**
>
> In our original submission, we have provided and compared the visualization results of correlation maps and images before the attack (i.e., clean examples), after the attack (adversarial examples), and after reconstruction (defense examples) in the supplementary material (i.e., Appendix A.3) to validate our performance. To address the concerns, we added more visualization results with higher resolutions in the supplementary material with Figure 7, Figure 8, and Figure 9. Moreover, we have added corresponding explanations in the main submission (Sec. 4.1).

---

> ### Author Response · Authors · 2023-11-22
> **Q3: For the VOT, does the proposed method reconstruct the whole search image?**
>
> Thank you for your comment. We have configured the reconstruction range to be the search region rather than the entire image, significantly reducing the time costs. This is also aligned with the existing trackers' pipeline. To avoid confusion, we have revised the `Other details.' subsection in Sec. 3.4.

---

> ### Author Response · Authors · 2023-11-22
> **Q4: The writing of the paper needs to improve and some results in supplementary materials can be combined in the main paper.**
>
> Thank you for your valuable suggestions. We've diligently proofread the entire submission, rectifying some typos and grammar errors. Due to space constraints, certain content is included in the supplementary material. We have integrated these details with our main paper as a whole, appropriately citing the results to ensure readers are aware of the relevant information in the supplementary material.

---

### Author Response · Authors · 2023-11-22
**Summary**

We appreciate the thoughtful comments and helpful criticism from every reviewer. We are delighted that the reviewers thought our paper was "reasonable," "interesting," "easy to follow," and "experiments are thorough."

Below is a summary of our paper update, and we marked the updates in our paper with blue color. More results could be found in the response.

1. We've diligently proofread the entire submission, rectifying some typos and grammar errors.
2. **[Experimental Results 6.1]** We have added the results of diffusion-based defense (i.e., DiffPure (Nie et al. 2022)) to Appendix A.7. Additionally, a discussion has been included in Sec. 2 for a more comprehensive coverage.
3. **[Appendix A.1 & Section 4.2]** We have added the results on TrackingNet to Table 7 in Appendix A.1 and explanations in Sec 4.2.
4. **[Section 3.4]** We have detailed the setups of the reconstruction range in Sec. 3.4 to avoid any confusion.
5. **[Appendix A.3 & Section 4.1]** We have added more visualization results in Appendix A.3 with Figure 7, Figure 8, and Figure 9. We have added corresponding explanations in Sec. 4.1.
6. **[Appendix A.8 & Section 4.2]** We have added the results of resizing and compression-based defense in Appendix A.8 with Table 12 and Table 13. We also cite these results in Sec. 4.2.
7. **[Appendix A.9 & Section 4]**  We have added a section in the supplementary material to detail the implementation of attacks with the explanation in Sec.4.
8. **[Section 3.3]** We have updated Section 3.3 by incorporating explanations on how to obtain the text from a given template.
9. **[Section 5]** We have incorporated a discussion in the conclusion section to delve into the attack and defense of nature-language-based tracking as part of future works.
10. **[Section 2]** We have updated the related work to involve the recent transformer-based trackers.

---

### Meta-Review · Area_Chair_hs1L · 2023-12-09

**Metareview:**

The paper initially had mixed reviews (5,6,6). The major concerns were:

1. Does the proposed method reconstruct the whole image? Could inversion methods in StyleGAN or Diffusion also work? [bxnS, n1Fj]
2. Show qualitative results between clean, adversarial, and defended images. [bxnS]
3. writing needs improvement. Some supplementary material should be combined with the main paper [bxnS]
4. missing comparisons with basic defense methods (adversarial training, image processing [n1Fj]
5. which attack method is used in Table 1? [n1Fj]
6. why is NLP embedding needed? A template seems sufficient. [DMWE]
7. what if the object class is unknown? [DMWE]
8. missing experiments on more challenging benchmarks, tracking net and tnl2k [DMWE]
9. only SiamRPN++ is used as the baseline, which is out of date. Does the method work with transformer models, OSTrack, SwinTrack, MixFormer? [DMWE]
10. What if the tracking method already uses NLP? [DMWE]

The authors wrote a response to address the concerns. Specifically, they provided more comparisons with diffusion models and simple baselines, more details on the methodology and examples. Authors also provided additional experiments on larger benchmarks, and with a transformer-based tracker. The negative reviewer still thought that the novelty of introducing DISCO-like defense to visual tracking was limited, but wouldn't mind if the paper was accepted. The other reviewers did not change their ratings.

Overall, the AC thinks the novelty is sufficient, since the idea of DISCO is used but other methodology is required for tracking (based on CLIP) . Indeed the proposed work improves over the vanilla DISCO in the experiments. The concerns of the reviewers were addressed well, and all reviewers thought that using the language-image model as guidance was interesting, and the experiments were well done and good. Thus the AC recommends accept. The authors should prepare a camera-ready version based on the reviews, response, and discussion.

**Justification For Why Not Higher Score:**

- the paper adapts an image-based defense method DISCO to video tracking, so it is not a significantly novel idea.

**Justification For Why Not Lower Score:**

- the results are good, and outperform the image-based baseline DISCO.
- there is some novelty in how they design the implicit spatio-temporal function (as opposed to a spatial function).

---

### Decision · Program_Chairs · 2024-01-16

Accept (poster)